# Learning differentially shapes prefrontal and hippocampal activity during classical conditioning

Jan L Klee*, Bryan C Souza, Francesco P Battaglia*

Donders Institute for Brain, Cognition and Behaviour, Radboud University, Nijmegen, Netherlands

**Abstract** The ability to use sensory cues to inform goal-directed actions is a critical component of behavior. To study how sounds guide anticipatory licking during classical conditioning, we employed high-density electrophysiological recordings from the hippocampal CA1 area and the prefrontal cortex (PFC) in mice. CA1 and PFC neurons undergo distinct learning-dependent changes at the single-cell level and maintain representations of cue identity at the population level. In addition, reactivation of task-related neuronal assemblies during hippocampal awake Sharp-Wave Ripples (aSWRs) changed within individual sessions in CA1 and over the course of multiple sessions in PFC. Despite both areas being highly engaged and synchronized during the task, we found no evidence for coordinated single cell or assembly activity during conditioning trials or aSWR. Taken together, our findings support the notion that persistent firing and reactivation of task-related neural activity patterns in CA1 and PFC support learning during classical conditioning.

## Introduction

The ability to react to sensory cues with appropriate behavior is crucial for survival. On the level of neuronal circuits, linking cues to actions likely requires the interplay between a large network of cortical and subcortical brain structures, including the medial prefrontal cortex (PFC) and the CA1 area of the hippocampus (*Allen et al., 2017*; *Steinmetz et al., 2019*). Both areas have been found to respond to sensory cues and reward in various behavioral paradigms (*Aronov et al., 2017*; *Chen et al., 2013*; *Starkweather et al., 2018*; *Taxidis et al., 2020*) and are involved in action planning and execution (*Otis et al., 2017*; *Terada et al., 2017*).

PFC has been suggested to map contextual and sensory information to appropriate actions according to flexible rules (*Euston and Gruber, 2012*). Accordingly, PFC has been found to control the development and expression of anticipatory licking during sensory guided reward-seeking behavior (*Otis et al., 2017*). PFC also maintains working memory representations of sensory cues over delay periods (*Funahashi et al., 1993*; *Goldman-Rakic, 1995*).

Similar to PFC, CA1 responds to sensory cues and displays sustained activity during delay periods to support memory formation (*Hattori et al., 2015*; *McEchron et al., 1999*; *McEchron and Disterhoft, 1997*).

Importantly, CA1 and PFC interact substantially during awake hippocampal Sharp-Wave Ripples (aSWRs) (*Jadhav et al., 2016*). aSWRs have been suggested to support planning of goal-directed actions in the context of spatial navigation (*Ólafsdóttir et al., 2018*) and the disruption of aSWRs leads to impairments in anticipatory behavior (*Nokia et al., 2012*). In addition, sensory cue representations are reactivated in hippocampal and corticohippocampal circuits during aSWRs (*Herzog et al., 2020*; *Rothschild et al., 2017*). Yet, whether task-related information during classical conditioning is also reactivated in the CA1–PFC circuit during aSWR and how this changes over the course of learning is currently unknown.

*For correspondence:
janlukasklee@gmail.com (JLK);
fpbattaglia@gmail.com (FPB)

Competing interest: The authors declare that no competing interests exist.

Here, we investigate how neural activity patterns in CA1 and PFC change throughout learning of sensory guided behavior, if information related to sensory cues is maintained while anticipatory actions are performed, and whether task-related information is reactivated in the CA1–PFC circuit during aSWRs. To this end, we employed high-density silicon probe recordings from both areas in head-fixed mice during appetitive auditory trace conditioning (AATC). Our findings reveal that CA1 and PFC exhibit distinct learning-dependent changes in sensory cue evoked activity, trial type-, and sensory-cue-related sustained activity as well as reactivation of task-related neural assemblies during aSWR.

## Results

### Head-fixed mice learn to anticipate reward during AATC

We trained head-fixed mice to associate one of two sounds (CS+ vs CS−) with a liquid reward delivered after a 1-s-long, silent trace period (*Figure 1A*; *Otis et al., 2017*). Successful learning expressed as the emergence of anticipatory licking of the animals in response to CS+ sounds (*Figure 1B and C*). Across all animals, lick rates during the trace period were significantly higher during CS+ trials after 5 days of training (*Figure 1C*; Wilcoxon rank sum, p < 0.01).

### CA1 and PFC exhibit learning-dependent changes in sound evoked and sustained activity

We next investigated how learning shapes neural dynamics in CA1 and PFC. To this end, we performed high-density silicon probe recordings from dorsal CA1 and PFC (1636 and 2217 cells total and 34 and 54 average per session in CA1 and PFC, respectively; *Figure 1—figure supplement 1* and *Figure 1—source data 1*), during the first 2 days of training and after the animals successfully acquired the conditioning task (hereafter referred to as pre- and postlearning) (*Figure 1D, E and J*). We observed that over the course of learning both areas developed pronounced differences in neural activity patterns between CS+ and CS− trials.

Short-latency evoked responses to both CS+ and CS− stimuli increased over the course of learning in CA1 (Kruskal–Wallis test; $F(3,3267) = 23.64$, p < 0.001; post hoc Wilcoxon rank sum test for CS+, p < 0.001 and CS−, p = 0.008) but were not significantly different from each other postlearning (*Figure 1G*, Wilcoxon rank sum, p = 0.73). In contrast, in PFC, CS+ responses remained high during learning but responses to CS− stimuli decreased (Kruskal–Wallis test; $F(3,4413) = 21.49$, p < 0.001; post hoc Wilcoxon rank sum test for CS+, p = 0.2 and CS−, p < 0.001) which led to significant differences in postlearning responses between CS+ and CS− stimuli (*Figure 1L*, Wilcoxon rank sum p < 0.05).

During the trace period following CS+ sounds, cells in PFC exhibited, on average, a strong sustained increase in firing rates (Wilcoxon rank sum, p < 0.001) (*Figure 1M*) while average single-cell responses in CA1 became significantly suppressed (Wilcoxon rank sum, p < 0.001) (*Figure 1H*).

Reward evoked activity significantly decreased in both CA1 (Wilcoxon rank sum, p < 0.001) and PFC (Wilcoxon rank sum, p < 0.001) (*Figure 1I and N*) from pre- to postlearning sessions.

### A subset of single cells in CA1 and PFC show lick-related activity

Over the course of learning, mice started to respond to CS+ sounds with anticipatory licking. Therefore, we investigated whether CA1 and PFC also exhibited time-locked activity at the onset of anticipatory licking during CS+ trials. In line with previous reports showing the involvement of PFC in licking behavior during appetitive trace conditioning (*Otis et al., 2017*), we found that single cells in PFC showed increases in activity compared to pretrial baseline at the time of the first anticipatory lick (PFC Lick-Up, n = 77, 4 % of all cells; criterion: mean activity during −250 to +250 ms around first lick, one standard deviation above pretrial baseline) (*Figure 2D*). We also found that a small population of CA1 cells responded during licking (CA1 Lick-Up, n = 36, 2%) (*Figure 2A*).

### CA1 and PFC cells exhibit distinct patterns of trial type-specific sustained activity

However, lick-related activity was not the only driver of single-cell modulation during the interval between CS+ and reward delivery. We discovered a large subset of single cells that exhibited trial

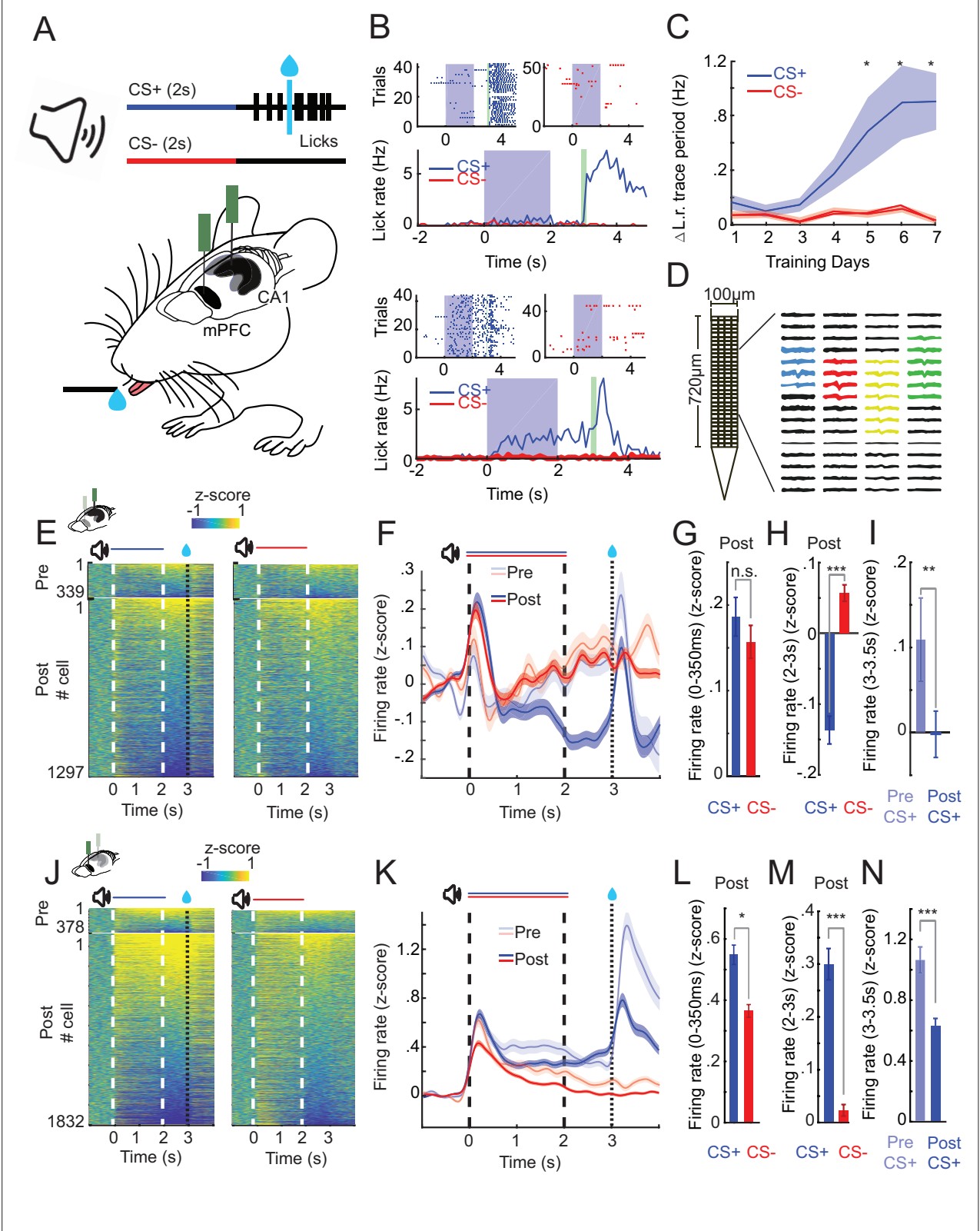

**Figure 1.** CA1 and prefrontal cortex (PFC) single-cell activity shows distinct learning-dependent changes during appetitive auditory trace conditioning (AATC). (**A**) Schematic of AATC task and electrophysiological recordings. (**B**) Example postlearning training sessions of one mouse during the AATC task (dots in raster plots represent licks, solid lines indicate average responses from respective sessions). (**C**) Average change in lick rate during the trace period trial during learning for all animals (*n* = 17) (* indicates sessions with significantly higher group average licks during the trace period after CS+

*Figure 1 continued on next page*

*Figure 1 continued*

sounds, Shade area represents standard error of the mean [SEM]). (**D**) 'Neuroseeker' silicon probe layout and combined spatial spike waveform patterns of four simultaneously recorded example neurons from CA1. (**E**) Z-scored firing rates of all CA1 neurons recorded pre- (top) and post (bottom) learning during CS+ and CS− trials ordered according to average trace period firing rates. (**F**) Z-scored Peri-Stimulus-Time-Histograms (PSTHs) of all recorded cells in CA1. (**G**) Z-scored sound evoked change in firing rate (0–350 ms post CS+/CS− onset) in CA1. (**H**) Z-scored trace period change in firing rate (2–3 s post CS+/CS− onset) in CA1. (**I**) Z-scored reward period change in firing rate (0–.5 s postreward presentation for CS+ trials pre- and postlearning) in CA1. (**J**) Z-scored firing rates of all PFC neurons recorded pre- (top) and post (bottom) learning during CS+ and CS− trials ordered according to average trace period firing rates. (**K**) Z-scored PSTHs of all recorded cells in PFC. (**L**) Z-scored sound evoked change in firing rate in PFC. (**M**) Z-scored trace period change in firing rate in PFC. (**N**) Z-scored reward period change in firing rate in PFC (*, **, and *** represent Wilcoxon rank sum, p < 0.05, p < 0.01, and p < 0.001) (error bars and shaded areas represent SEM).

The online version of this article includes the following figure supplement(s) for figure 1:

**Source data 1.** The number of recorded neurons per animal and session in CA1, prefrontal cortex (PFC), and simultaneous CA1–PFC recordings.

**Figure supplement 1.** Positioning of silicon probes in CA1 and prefrontal cortex (PFC).

---

type-specific sustained responses during the trace period in CA1 and PFC, similar to what had previously been described during aversive eyeblink trace conditioning (*Hattori et al., 2015*; *Hattori et al., 2014*; *Takehara-Nishiuchi and McNaughton, 2008*).

In CA1, a large fraction of nonlick cells was significantly suppressed by CS+ stimuli (post Trace-Down, *n* = 580, 40%) (*Figure 2B and C*, bottom) and the percentage of these Trace-Down cells as well as suppression levels increased from pre- to postlearning (Pre 19 %; Suppression Pre vs Post, *t*-test, p < 0.01). We also observed a smaller fraction of nonlick cells with sustained increases to CS+ sounds (post Trace-Up, *n* = 247, 17%) (*Figure 2B and C*, bottom). The percentages of these cells decreased from pre- to postlearning session (Pre, 29%, *Figure 3—figure supplement 1A* and B).

In PFC, most modulated cells showed sustained increases in activity in response to CS+ sounds (Trace-Up, *n* = 734, 38%) (*Figure 2E and F*, top) and the percentage and activation levels of these cells remained similar from pre- to postlearning. A similar fraction of cells showed sustained suppression (Trace-Down, *n* = 630, 33%) (*Figure 2E and F*, bottom). Differences between CS+ and CS− responses that emerged over the course of learning in PFC were mostly caused by a reduction in CS− stimulus evoked activity. CS− responsive Trace-Up cells in PFC significantly decreased their activity from pre- to postlearning sessions (*t*-test, p < 0.001) (*Figure 2—figure supplement 1A and B*).

## CA1 and PFC population activity distinguishes between rewarded and unrewarded trials during the trace period

Because individual cells in CA1 and PFC exhibited sustained activity during the trace period, we hypothesized that these responses might be part of a broader CA1 and PFC population code to maintain a representation of trial identity between CS+ and reward delivery (i.e., in the trace period, in which there is no ongoing stimulus). To test this, we first computed the binned population rate vectors for all simultaneously recorded nonlick cells in both areas during CS+ and CS− trials on a session-by-session basis. We next calculated the Euclidean distance between the population rate vector trajectories during the two stimuli (see *Figure 3A and C* for trajectory examples), and used this metric as a proxy for the similarity of population responses over time. We found that in postlearning sessions, the population rate vector distance increased after stimulus onset and persists to be significantly different from baseline during the trace period in CA1 (Wilcoxon sign rank, p < 0.001) and PFC (Wilcoxon sign rank, p < 0.001), indicating that both areas maintain trial type-specific information at the population level. Nonlick cell population rate vector distance did not correlate with lick activity or movement of the animals in CA1 (*n* = 36, Population Vector Distance vs Licks: correlation coefficient = 0.14, p = 0.42; Population Vector Distance vs movement: correlation coefficient = 0.03, p = 0.85; *Figure 3—figure supplement 2C*) or PFC (*n* = 38, Population Vector Distance vs Licks: correlation coefficient = −0.1, p = 0.52; Population Vector Distance vs movement: correlation coefficient = −0.04, p = 0.74; *Figure 3—figure supplement 2D*). Population response differences persisted even after reward delivery and usually settled back at baseline levels after 20 s (*Figure 3—figure supplement 1*).

To verify that both areas maintain CS type-specific information on a trial-by-trial basis, we next trained a support vector machine classifier on the firing rates of simultaneously recorded nonlick cells during the trace period from either CA1 or PFC. We were able to predict the preceding stimulus identity significantly above chance level in CA1 (mean performance, 62 % correct, Wilcoxon sign

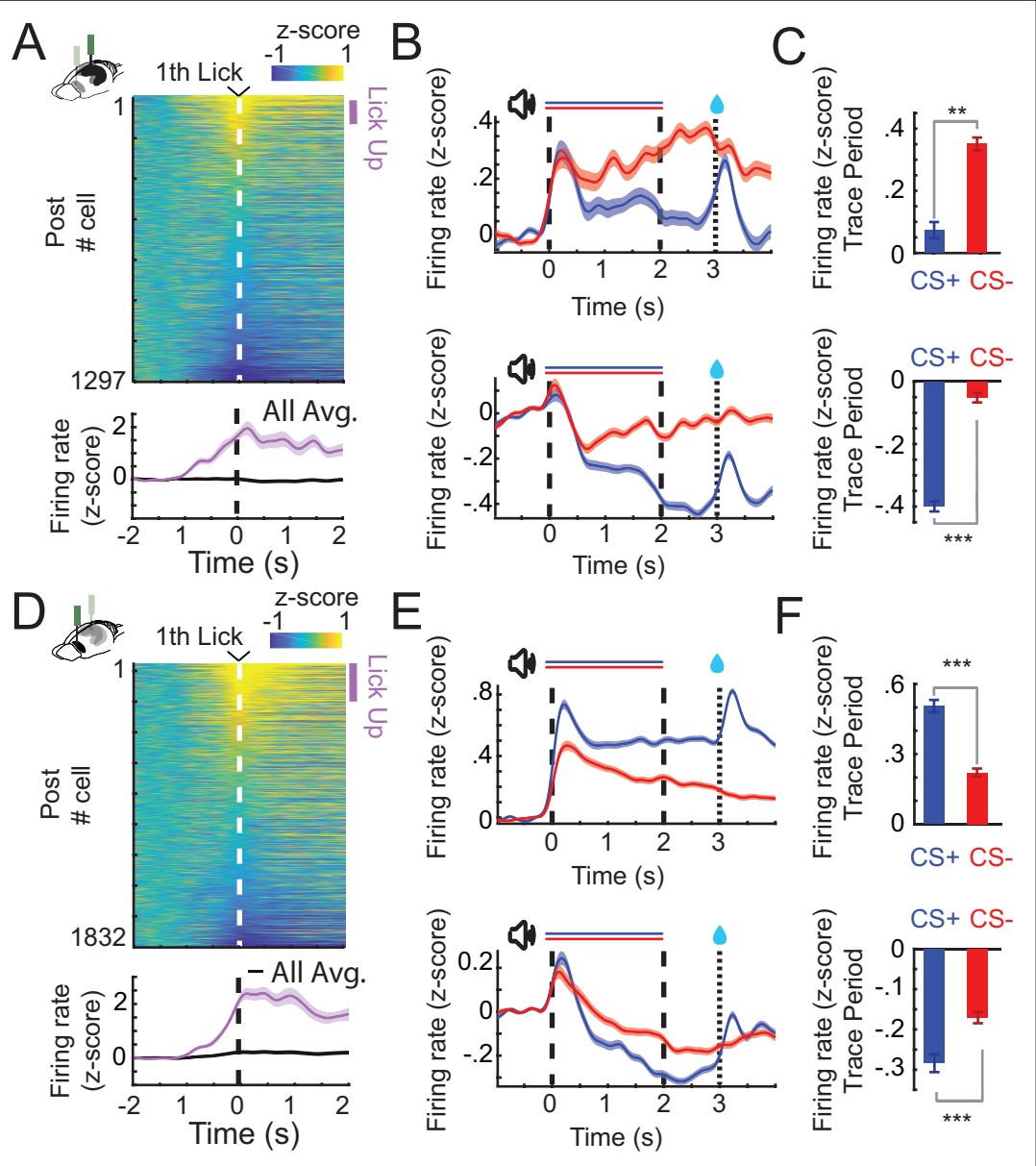

**Figure 2.** CA1 and prefrontal cortex (PFC) single cells exhibit lick evoked responses and distinct patterns of sustained activity. (**A**) Z-scored firing rates of all CA1 neurons (top) aligned to the first lick of a lick bout (at least three licks/s) during CS+ trials (before reward delivery). Z-scored change in activity of all and for positively lick modulated cells (bottom). Purple bar indicates Lick-Up cells. (**B**) Z-scored Peri-Stimulus-Time-Histograms (PSTHs) of all Trace-Up (top) and Trace-Down (bottom) postlearning for CA1 (in CS+ or CS− trials: Trace-Up, n = 444; Trace-Down, n = 675). (**C**) Z-scored change in firing rate during the trace period of the same Trace-Up neurons (top) and Trace-Down neurons (bottom) for CA1. (**D**) Lick cells in PFC (same as in A). (**E**) Trace-Up (top) and Trace-Down (bottom) nonlick neurons postlearning for PFC (CS+ or CS− trials: Trace-Up, n = 736; Trace-Down, n = 734). (**F**) Z-scored change in firing rate during the trace period of the same Trace-Up neurons (top) and Trace-Down neurons (bottom) for PFC (**, and *** represent Wilcoxon rank sum, p < 0.01, and p < 0.001; error bars and shaded areas represent standard error of the mean [SEM]).

The online version of this article includes the following figure supplement(s) for figure 2:

**Figure supplement 1.** Distribution and activation of Trace-Up and Trace-Down cells in CA1 and prefrontal cortex (PFC) changes over the course of learning.

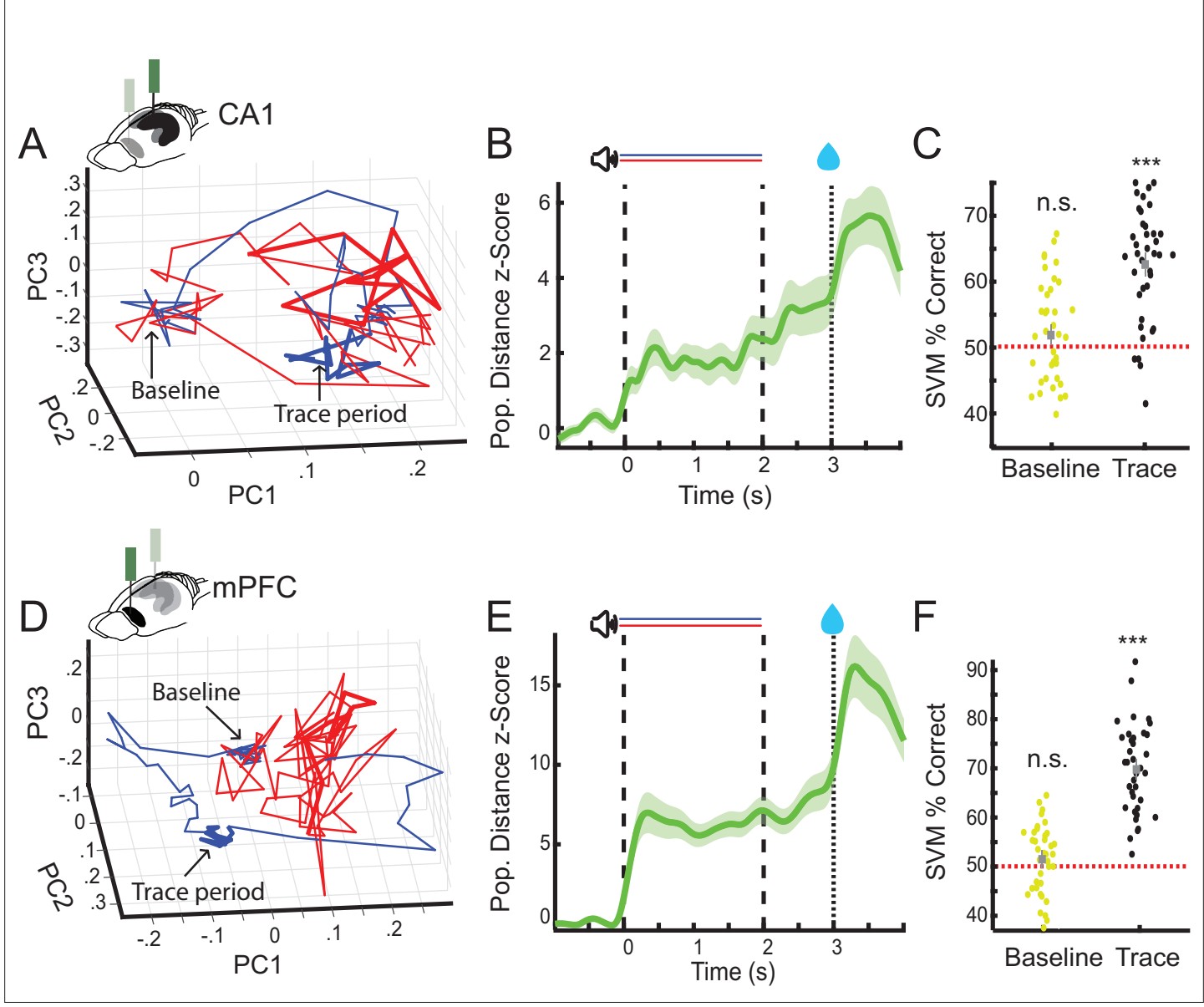

**Figure 3.** CA1 and prefrontal cortex (PFC) nonlick cell population activity encode trial identity during the trace period. (**A**) Example of average nonlick cell population rate vector trajectories for one session in CA1 (CS+ [blue] and CS− [red]). Averages plotted along first three principal components (baseline period: blob on the left, trace period thicker lines on the right). (**B**) Average z-scored Euclidean distance between CS+ and CS− nonlick cell population rate vector trajectories during appetitive auditory trace conditioning (AATC) task for CA1 (*n* = 36) (shaded areas represent standard error of the mean [SEM]). (**C**) Support vector machine classification of trial identity by average baseline (−1 s-0 s) and trace period (2–3 s) activity of nonlick cells in CA1 (*n* = 36) (*** indicates Wilcoxon sign rank p < 0.001). (**D**) Example of average nonlick cell population rate vector trajectory for one session in PFC. (**E**) Average z-scored Euclidean distance between CS+ and CS− nonlick cell population rate vector trajectories for PFC (*n* = 38) (shaded areas represent SEM). (**F**) Support vector machine classification of trial identity by average baseline (−1 s-0) and trace period (2–3 s) activity of nonlick cells in PFC (*n* = 38).

The online version of this article includes the following figure supplement(s) for figure 3:

**Figure supplement 1.** CA1 and prefrontal cortex (PFC) single cells and population responses slowly decay back to baseline after conditioning trials.

**Figure supplement 2.** CA1 and prefrontal cortex (PFC) nonlick cell population activity does not correlate with lick or running behavior.

rank p < 0.001; *Figure 3C*) and PFC (mean performance, 76 % correct, Wilcoxon sign rank p < 0.001; *Figure 3F*).

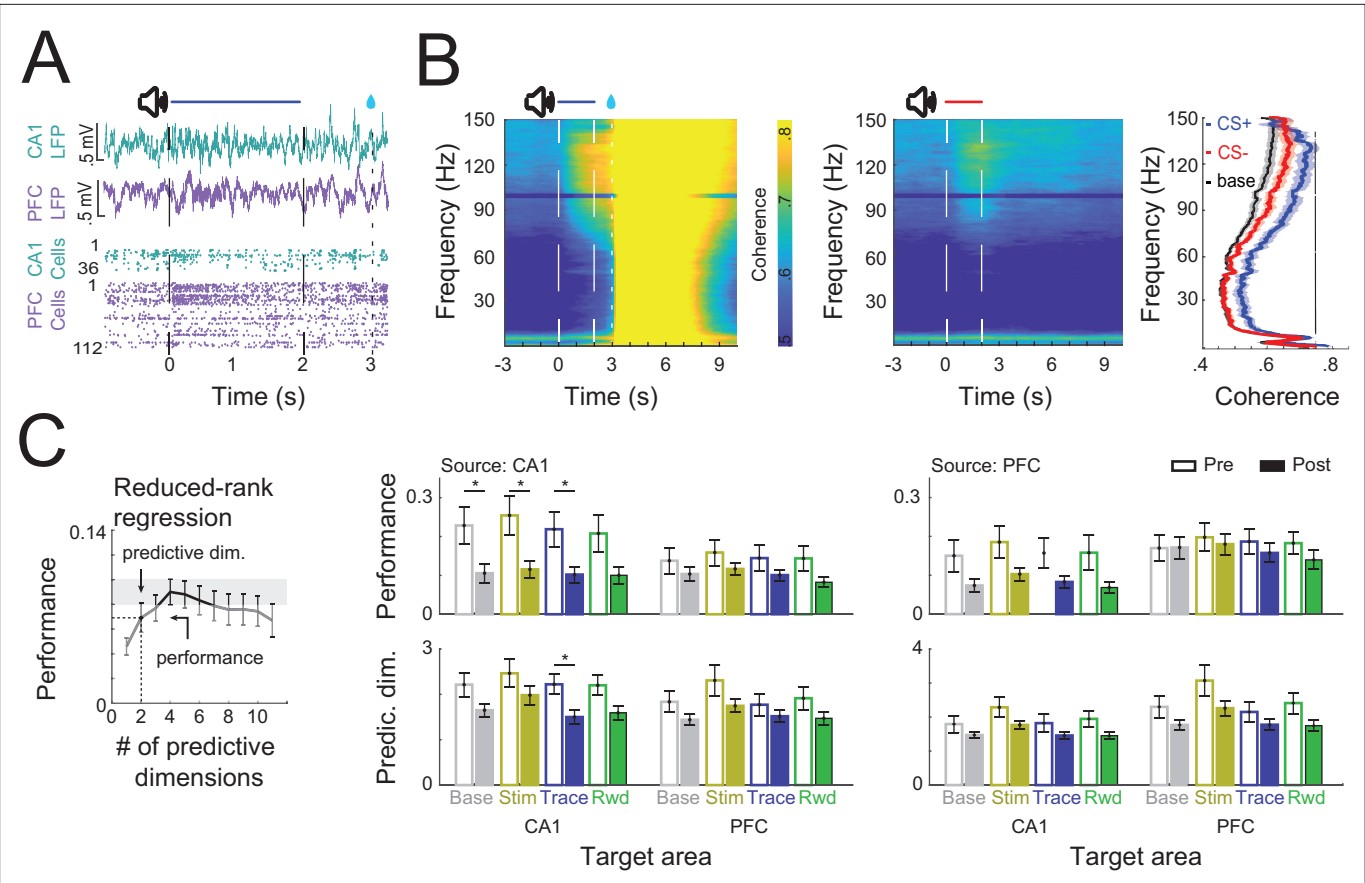

**Figure 4.** CA1–PFC interaction during trace conditioning. (**A**) Example of simultaneously recorded local field potential (LFP) and single-cell activity from CA1 and PFC during a CS+ conditioning trial. (**B**) CA1–PFC LFP coherence during CS+ trials (right) and CS− trials (left). Average coherence during baseline and during the trace period (right). Black bar indicates significant difference between CS+ post and CS− post trials (permutation test at each frequency <0.05). (**C**) (Left) Schematic representation of how performance and the number of predictive dimensions were calculated for each regression. (Right) Reduced-rank regression between CA1 and PFC spiking activity during conditioning trials in pre- and postlearning sessions (Solid and filled bars represent prelearning and postlearning sessions, respectively, * refers to Wilcoxon ranksum test p<0.05, error bars represent standard error of the mean [SEM]).

The online version of this article includes the following figure supplement(s) for figure 4:

**Figure supplement 1.** CA1–prefrontal cortex (PFC) single-cell interaction does not change across different task periods.

## CA1–PFC local field potential coherence but not single-cell interactions increase during trace conditioning

CA1 and PFC are known to interact during various spatial memory tasks in rodents (*Benchenane et al., 2010*; *Jones and Wilson, 2005*; *Sigurdsson et al., 2010*; *Spellman et al., 2015*) and aversive trace conditioning affects local field potential synchronization within and across brain areas (*Shearkhani and Takehara-Nishiuchi, 2013*; *Takehara-Nishiuchi et al., 2011*). Given that CA1 and PFC are also highly engaged and encode information about trial identity during appetitive trace conditioning, we wondered if we could find evidence for an interaction between CA1 and PFC on the level of single cells and local field potentials (LFPs) (*Figure 4A*).

We found that postlearning, high-frequency CA1–PFC LFP coherence was increased during CS presentation (n = 13, 60–140 Hz permutation test at each frequency <0.05) (*Figure 4B*). During CS+ trials, trace period coherence was furthermore significantly elevated across a broad frequency range (n = 13, 12–140 Hz permutation test at each frequency <0.05).

Given the interactions between CA1 and PFC on the LFP level, we next checked for an interaction between both areas on the single-cell level (*Figure 4C*). To this end, we computed a reduced-rank regression (RRR) to assess how well the activity of a sampled population in one of the areas

(source area) could be used to explain another (disjoint) sampled population in the same area or in another connected area (target area), through a simplified, low-dimensional linear model. We then used crossvalidation to estimate the optimal dimensionality (rank) of each RRR and its performance ($R^2$) (*Semedo et al., 2019*).

We observed that postlearning CA1 ensemble activity at baseline could be used to predict other individual neurons firing rates in CA1 just as well as the firing rates of neurons in PFC. The PFC ensemble, on the other hand, was much better at predicting the firing rates of other PFC neurons compared to neurons in CA1 (*Figure 4C*), which indicates a directionality of information flow between both areas at baseline.

However, crossarea predictability of firing rates did not change significantly when we compared baseline levels to any of the different trial stages (Stim, Trace, or Reward) (*Figure 4C*, Wilcoxon rank sum test; p < 0.05). Comparing the performance of the full-rank model also did not reveal any significant differences in coordination between CA1 and PFC across task periods (ridge regression with L1 regularization; *Figure 4—figure supplement 1*, Wilcoxon rank sum test; p < 0.05).

However, by focusing on CA1, we found that ensemble activity substantially decorrelated over the course of learning and individual cells firing rates were significantly less well predicted by the rest of the ensemble from pre- to postlearning sessions. This was not the case in PFC (*Figure 4C*).

## Task-related neuronal assemblies are more strongly reactivated in PFC during aSWR after learning

Learning-dependent reorganization of cortical circuits during memory consolidation has previously been linked to activity during hippocampal SWRs (*Peyrache et al., 2009*) and reactivation of spatial information in PFC during aSWRs has been reported by several groups (*Kaefer et al., 2020*; *Maggi et al., 2018*; *Shin et al., 2019*). aSWRs have additionally been implicated in the planning of goal-directed behavior (*Ólafsdóttir et al., 2018*). Therefore, we wondered if we could find evidence for reactivation of task-related neural assemblies during aSWRs occurring during intertrial intervals of the conditioning task. To test this, we first detected the presence of neuronal cell assemblies in concatenated trial activity (*Lopes-dos-Santos et al., 2013*, *Figure 5—figure supplement 3*) and then checked the reactivation strength of these task-related assemblies during aSWRs in CA1 and PFC. We found that reactivation of task-related assemblies in PFC increases significantly over the course of learning during hippocampal aSWRs (*Figure 5A and B*) (Wilcoxon rank sum test; p < 0.05). This was true for assemblies defined during trials as well as during intertrial intervals (*Figure 5—figure supplement 1*). In CA1, on the other hand, the reactivation strength remained constant from pre- to postlearning sessions (p = 0.337). The frequency of aSWR occurrences did not change between pre- and postlearning sessions (Pre *n* = 24, 0.08 Hz; Post *n* = 38, 0.09 Hz; Wilcoxon rank sum test; p = 0.2) (*Figure 5—figure supplement 2B, C*). During conditioning trials, aSWR rate decreased and was at its lowest during reward consumption (*Figure 5—figure supplement 2C*). Average aSWR rates independent of task stage slightly increased from the beginning to the end of each session (*n* = 62; 0.06–0.1 Hz, Wilcoxon rank sum test; p < 0.01).

## Task-related assembly reactivation strength increases during individual sessions in CA1

We next analyzed how reactivation strength changed over time within individual sessions. We found that in CA1 assembly reactivation strength per aSWR increases gradually over the course of individual prelearning sessions for positively modulated assemblies (reactivation+) as well as for negatively modulated assemblies (reactivation−) during aSWRs (*Figure 5C*). This effect was also present in postlearning sessions for the reactivation+ assemblies (one-way ANOVA; CA1 pre+: p < 0.01, pre−: p < 0.05, post+: p < 0.001, post−: n.s.). In PFC assembly reactivation strength slightly increased for positively modulated assemblies in postlearning sessions (*Figure 5C*).

We then sought to determine what type of task-specific information the most strongly SWR-reactivated assemblies represent during the conditioning task in postlearning sessions. In CA1, we found that the 25 % most reactivated assemblies are suppressed during the trace and reward period after CS+ trials compared to CS− trials (*Figure 5D*) (Wilcoxon signed-rank; p < 0.001). In PFC, on the other hand, the 25 % most reactivated assemblies responded strongly during the reward period

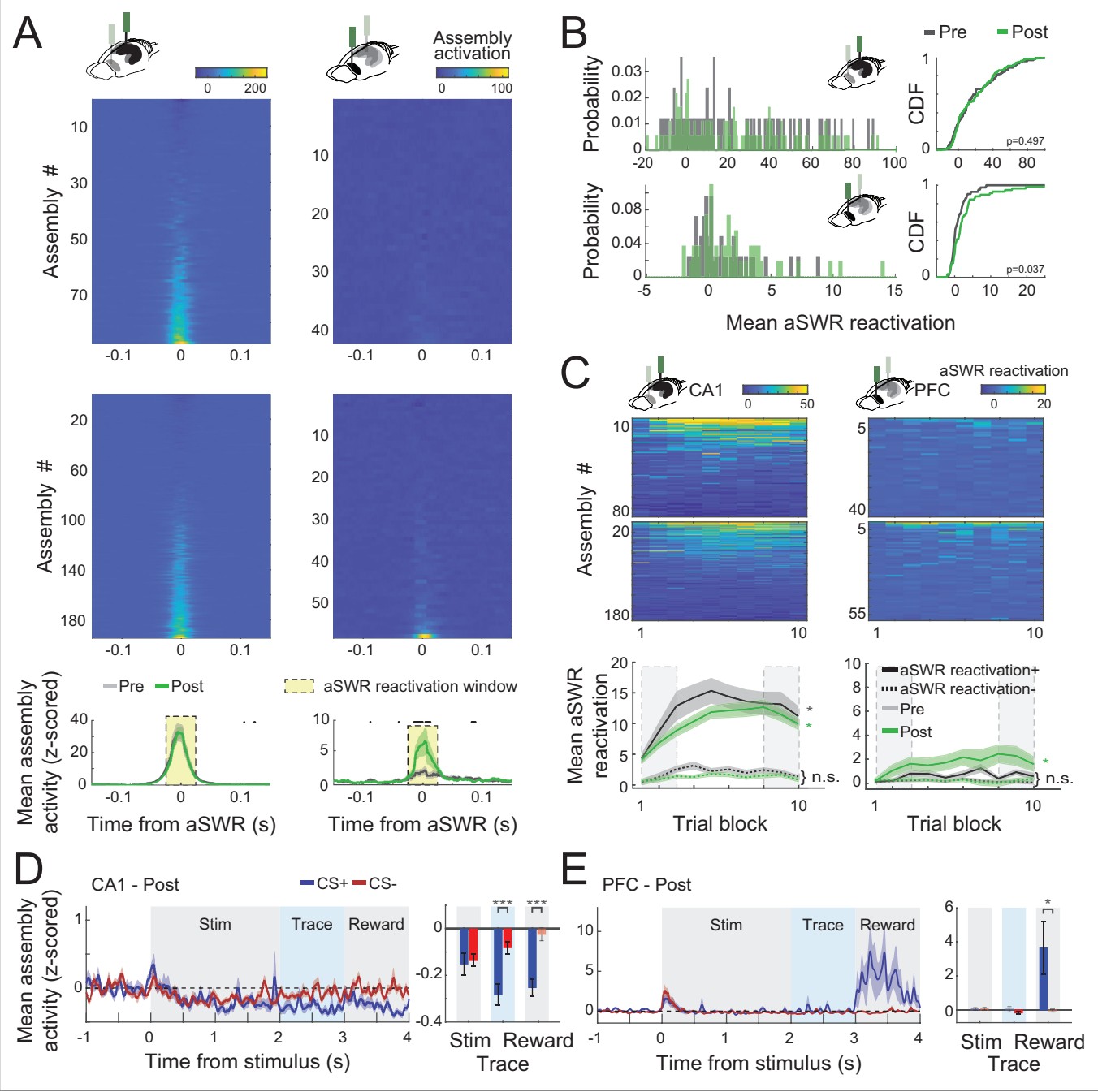

**Figure 5.** CA1 and prefrontal cortex (PFC) cell assemblies show different awake Sharp-Wave Ripple (aSWR) reactivation dynamics. (**A**) Average (z-scored) assembly activation triggered by aSWR occurring in the intertrial intervals for CA1 and PFC, pre- and postlearning sessions (top). Mean aSWR-triggered activation over all the assemblies for pre- and postsessions for each area. Shaded areas represent the standard error of the mean (SEM). Black dots represent windows in which pre- and postassembly activity were statistically different (Wilcoxon rank sum test; p < 0.05). Notice the higher aSWR-triggered activation of assemblies in PFC in postsessions. (**B**) Histogram (left) and cumulative distribution function (CDF; right) of the mean assembly activity on the reactivation window denoted in A. p values refer to a two-sample Kolmogorov–Smirnov test between pre- and postdistributions. (**C**) Average aSWR reactivation of each assembly per session (top). Sessions were divided into 10 blocks of equal trial length. Mean aSWR reactivation of all positively (reactivation+) and negatively (reactivation−) reactivated assemblies. Asterisks refer to Wilcoxon signed-rank test performed between the first and last three trial blocks (dashed rectangles) of each area/learning condition (n.s.: nonsignificant; *p < 0.05 and shaded areas represent SEM). Note the evident increase in CA1 aSWR assembly reactivation across the session in both pre- and postsessions for positively modulated assemblies (reactivation+). (**D**) Mean (z-scored) assembly activity triggered by the stimulus onset for the 25 % most strongly aSWR-reactivated assemblies in CA1 (left). Average of the traces over each trial period is shown for CS+ and CS− (right). Notice the initial decrease of assembly activity in CA1 during the stimulus and the posterior separation between CS+ and CS−. (**E**) The same as in D, but for PFC assemblies. Note the difference between CS+ and CS−

*Figure 5 continued on next page*

*Figure 5 continued*

assembly activity during the reward period. Asterisks refer to a Wilcoxon signed-rank test comparing CS+ and CS− (*p < 0.05; ***p < 0.001). Error bars refer to SEM and darker bars denote mean assembly activity significantly different from 0 (p < 0.05; t-test).

The online version of this article includes the following figure supplement(s) for figure 5:

**Figure supplement 1.** Awake Sharp-Wave Ripple (aSWR) reactivation of assemblies detected during intertrial intervals.

**Figure supplement 2.** Distribution of awake Sharp-Wave Ripples (aSWRs) during trace conditioning.

**Figure supplement 3.** Detecting cell assemblies in neural populations.

(*Figure 5E*) (*t*-test, compared to pretrial baseline; p < 0.05). Those effects were not observed for the 25 % least reactivated cells.

## CS+ responsive assemblies are preferentially replayed in CA1 during aSWRs

Given that assemblies detected during the task and the intertrial intervals became both reactivated during aSWR, we next asked whether we could find additional evidence for a prioritized reactivation of assemblies that carry specific information about CS sounds during the conditioning trials. To this end, we first computed the average activity of each assembly for CS+ or CS− stimuli and computed a trial-type modulation score, defined as the average assembly activation during CS+ trials subtracted by the average activation during CS− trials (stimulus and trace periods; *Figure 6A*). Then, for each session, we selected the most positively modulated (i.e., CS+ activity higher than CS−), the most negatively modulated (i.e., CS− activity higher than CS+), and the least modulated, control assembly. Because we found that most assemblies in CA1 are suppressed during the stimulus (as are the firing rates), we term assemblies by whether they were more suppressed during CS+ or CS− stimuli. We found that, for CA1, CS+ suppressed assemblies in both stimulus and trace periods (i.e., assemblies that are more suppressed during CS+ trials than CS− trials) were more reactivated during aSWRs (*Figure 6B*) compared to CS− suppressed assemblies and control, nonmodulated assemblies. This effect was confirmed by the presence of a negative correlation between the trial-type modulation score and the average aSWR reactivation of each assembly (*Figure 6C*).

CS− suppressed assemblies were also more strongly reactivated than control assemblies in prelearning sessions (p < 0.05, *Figure 6B*) and, in CS− trials, assemblies that were more suppressed during stimulus compared to trace period also reactivated more during aSWR (*Figure 6—figure supplement 1C*). Together this indicates that stimulus-coding assemblies are also differentially modulated during aSWR.

In PFC, we did not observe preferential reactivation of CS+ or CS− specific assemblies (*Figure 6—figure supplement 1*).

Finally, we wondered whether CA1 and PFC assembly reactivation is coordinated during aSWRs. Coordinated reactivation of task relevant information during aSWR has previously been found in the CA1–PFC circuit during some spatial navigation tasks but not during others (*Kaefer et al., 2020*; *Shin et al., 2019*). We found that both areas independently of each other showed comparable and significant aSWRs reactivations rates (*Figure 6—figure supplement 1D*). To then check whether CA1 and PFC assembly reactivation is coordinated during aSWR during trace conditioning, we computed how often high aSWR reactivation of pairs of assemblies cooccurred between the two areas (*Figure 6D*). We found that coordinated reactivations of pairs of CA1 and PFC assemblies during aSWRs happens at chance levels (*Figure 6D*) which suggest that aSWR reactivations of cell assemblies derived from conditioning trials are uncoordinated between CA1 and PFC.

## Discussion

This study characterizes changes in neural activity in the CA1–PFC network while mice learn to use predictive sounds to anticipate future rewards. We show that activity in both areas is strongly shaped by learning and that task-specific information is reactivated in a complex pattern across CA1 and PFC during aSWR.

While CA1 and PFC are highly active during aversive eyeblink trace conditioning, evidence for a similar involvement during appetitive trace conditioning had been missing. In fact, several previous

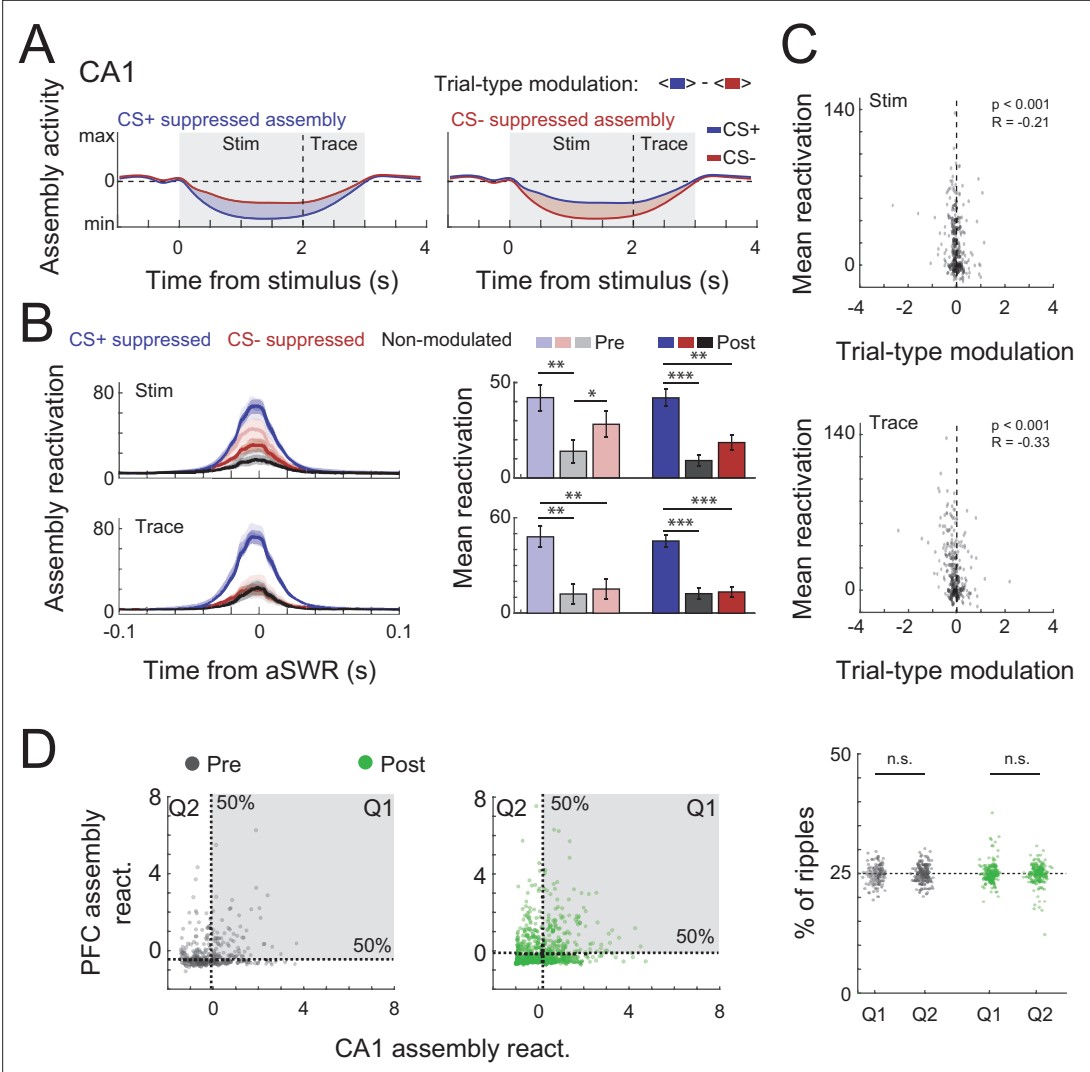

**Figure 6.** Trial-type modulation and prefrontal cortex (PFC) coactivation of CA1 assemblies. (**A**) Schematic representation of trial-type modulation scores, CS+ and CS− suppressed assemblies. The modulation score was defined as the difference between average assembly activation on CS+ and CS− trials during a specific period. (**B**) Mean awake Sharp-Wave Ripple (aSWR) reactivation of CS+ suppressed, CS− suppressed, and nonmodulated assemblies pre- and postlearning over time (left) and within 50 ms window around ripples (right). Error bars denote standard error of the mean (SEM) (*p < 0.05; **p < 0.01; ***p < 0.001). (**C**) Scatter plot and Pearson's correlation values between trial-type modulation score and average aSWR reactivation for all CA1 assemblies (pre- and postlearning). Notice the stronger reactivation of negatively modulated assemblies (CS+ suppressed). (**D**) (Left) Example of joint reactivation for two pairs of CA1–PFC assemblies. Quadrants were defined using the median aSWRs reactivation of each area and the proportion of reactivations in each quadrant was computed. (Right) Percentage of ripple reactivations in first and second quadrants defined in left for all possible combination of assembly pairs (Wilcoxon signed-rank test).

The online version of this article includes the following figure supplement(s) for figure 6:

**Figure supplement 1.** Trial-type modulation of prefrontal cortex (PFC) assemblies and trial-period modulation.

studies have pointed out differences in the mechanisms underlying both types of learning (*Pezze et al., 2017*; *Thibaudeau et al., 2007*).

Despite these differences, we found that single cells in CA1 and mPFC during appetitive trace conditioning behave similar to what had previously been reported during aversive trace conditioning. Both areas display long-lasting sustained activity that bridges the temporal gap between CS+ offset and reward delivery. In both areas, these sustained responses are composed of a mix of Trace-Up and Trace-Down cells, that is, cells that display sustained excitation and inhibition, respectively. In CA1, higher numbers and stronger inhibition of Trace-Down cells result in overall suppression

of the entire area during the trace period, while in PFC higher numbers and stronger activation of Trace-Up cells resulted in overall excitation.

Similar to our study, abundant Trace-Down like responses and sparse Trace-Up like responses have been described during aversive trace conditioning in CA1 (*Hattori et al., 2015*; *McEchron and Disterhoft, 1997*). This distinct pattern of mostly inhibition mixed with sparse excitation has been hypothesized to increase the signal-to-noise ratio to more efficiently propagate the signal of Trace-Up cells to downstream areas (*Hattori et al., 2015*). Yet, it is also conceivable that Trace-Down cells participate in an independent form of coding. Inhibition in CA1 might, for example, play an active role in suppressing well expected incoming stimuli, that is, reward delivery (*Bastos et al., 2012*; *Rummell et al., 2016*; *Stachenfeld et al., 2017*).

In PFC, responses during appetitive trace conditioning are also similar to what has previously been found during aversive trace conditioning. Specifically, higher numbers and stronger excitation of Trace-Up cells have also been found in rat PFC and parts of rabbit PFC during aversive trace conditioning (*Hattori et al., 2014*; *Takehara-Nishiuchi and McNaughton, 2008*). A learning-dependent reduction in responses to CS− like pseudoconditioning stimuli have also previously been described in PFC (*Hattori et al., 2014*; *Takehara-Nishiuchi and McNaughton, 2008*; *Weiss and Disterhoft, 2011*).

In combination, this suggests that sparse excitation with strong surrounding inhibition in CA1 and mostly excitation in PFC are two general coding principles employed to bridge the temporal gap between a salient cue and a behaviorally relevant event, independently of the appetitive or aversive nature of the event and the specific anticipatory action that it requires.

We furthermore observed high-frequency CA1–PFC coherence was specifically increased during CS+ trials. Increased synchronization between both areas during spatial working memory tasks have previously been reported predominantly in the theta frequency (4–12 Hz) (*Battaglia et al., 2011*; *Benchenane et al., 2010*; *Jones and Wilson, 2005*). However, during trace-conditioning mice were mostly stationary and did not display strong hippocampal theta oscillations and high-frequency coherence therefore likely results from a different underlying mechanism.

Despite both areas being highly engaged in the task and encoding trial-specific information on a trial-by-trial level, we did not find any evidence for task-specific communication on the single-cell level. CA1 and PFC therefore either process conditioning trials in parallel rather than in series or rely on intermediate structures (e.g., entorhinal cortex) for effective communication (*Insel and Takehara-Nishiuchi, 2013*).

Lastly, we found that cell assemblies in CA1 and PFC that are responsive during classical conditioning also strongly reactivate during hippocampal aSWRs. Earlier reports found coordinated aSWR reactivation of behavioral sequences during learning of spatial memory tasks in CA1 and PFC, respectively (*Kaefer et al., 2020*; *Shin et al., 2019*; *Shin and Jadhav, 2016*; *Tang et al., 2017*). In addition, close loop disruption of aSWRs has been shown to impair performance in spatial memory task and slow learning during conditioning. However, physiological evidence for reactivation of neural assembly patterns during aSWRs in a nonspatial task in either CA1 or PFC had been missing (*Joo and Frank, 2018*; *Ólafsdóttir et al., 2018*). Our data now provide this missing link.

Notably, we observed a fast increase in reactivation strength of assemblies derived from conditioning trial activity within CA1 during aSWRs as individual training sessions progressed and a slow increase in reactivation strength in trial and intertrial interval derived assemblies in PFC from prelearning to postlearning sessions. This is well in line with the idea that both areas support learning on different timescales and levels of complexity, with the hippocampus adapting fast to detailed new experiences and the PFC adapting more slowly to behaviorally important variables that remain stable over time (*McClelland et al., 1995*; *Takehara-Nishiuchi and McNaughton, 2008*; *Takehara et al., 2003*). Assembly reactivation in CA1 was furthermore task specific during trace conditioning. Assemblies that were suppressed during and after CS+ sounds became most strongly reactivated. CS− suppressed assemblies were also but less strongly modulated during aSWRs in CA1.

This preferential reactivation of CS suppressed assemblies during aSWR, provides additional evidence that suppression in CA1 during CS+ trials plays a pivotal role during trace conditioning and might be relevant to actively encode stimulus identity or to predict upcoming task events.

Surprisingly, we did not observe that assemblies, derived from activity during conditioning trials in CA1 and PFC, significantly coreactivated during aSWR.

Several studies previously reported single cells in CA1 and PFC with similar spatial firing fields to also be strongly correlated during aSWRs and that synaptic inputs to individual mPFC cells increased if CA1 replay was more coordinated (*Nishimura et al., 2021*; *Shin et al., 2019*; *Tang et al., 2017*). However, on the population level, CA1 and mPFC reactivation of specific spatial trajectories has been found to occur independently (*Kaefer et al., 2020*). Moreover, reactivation of spatial sequences in CA1 and other cortical areas, specifically the entorhinal cortex has been shown to occur independently as well (*O'Neill et al., 2017*).

One way to reconcile these findings is that the coordination of aSWRs reactivation within the CA1–PFC circuit might depend on task structure. If animals have to follow rules that are based on specific sequences of behavioral events, for example, in spatial alternation tasks (*Shin et al., 2019*), replay of sequences of events in CA1 might drive the activation of cells or cell assemblies in PFC that encode the appropriate behavioral response to those sequences (*Buzsáki and Tingley, 2018*). If the task structure instead 'only' requires stimulus response mappings, as in our experiment and visually guided spatial navigation experiments (*Kaefer et al., 2020*), PFC might not rely on additional information from CA1 and reactivation remains independent. However, coordinated aSWRs reactivation in CA1 and PFC might happen robustly in nonspatial tasks if the tasks require the animals to learn sequences (*Cabral et al., 2014*; *Rondi-Reig et al., 2001*; *Terada et al., 2017*).

Lastly, it would be intriguing to know how SWRs reactivation of task relevant information during classical conditioning depends on the current state of the animal. A previous study reported significant differences in coordinated CA1–PFC reactivation between wakefulness and sleep (*Tang et al., 2021*). Yet evidence for sleep SWR reactivation of nonspatial information is still lacking. This further highlights the importance to study SWR reactivation with a battery of different behavioral task and across behavioral states that can help to disentangle the exact content and relevance of replay events for learning and behavior in the future.

# Materials and methods

## Animals

For this experiment, we used a total of 17 male C57Bl/6 mice. The animals were obtained at 10–13 months of age from Charles River Laboratory and all experiments were performed within 2 months after delivery. Seven animals were used for silicon probe recordings from dorsal hippocampus area CA1, six animals were used for recordings from PFC, and an additional four animals were used for combined silicon probe recordings from CA1 and PFC during the AATC task. All animals were group housed until the first surgery after which they were individually housed to prevent damage to the implants. Throughout the experiment the animals were maintained on a reversed 12 h light/dark cycle and received food and water ad libitum until we introduced food restriction 2 days after the first surgery. All experiments were performed during the dark period. This study was approved by the Central Commissie Dierproeven (CCD) and conducted in accordance with the Experiments on Animals Act and the European Directive 2010/63/EU on animal research.

## Surgical preparation for head-fixed electrophysiological recordings

Animals were anesthetized using isoflurane (1–2%) and placed in a stereotaxic frame. At the onset of anesthesia, all mice received subcutaneous injections of carprofen (5 mg/kg) as well as a subcutaneous lidocaine injection through the scalp. The animals' temperature was maintained for the duration of the surgical procedure using a heating blanket. Anesthesia levels were monitored throughout the surgery and the concentration of isoflurane was adjusted so that a breathing rate was kept constant at around 1.8 Hz. We exposed the skull and inserted a skull screw over the cerebellum to serve as combined reference and ground for electrophysiological recordings. We then placed a custom made, circular head plate for head fixation evenly on the skull and fixated it with dental cement (Super-Bond C&B). For CA1 recordings, a craniotomy was performed over the left hippocampus –2.3 mm posterior and +1.5 mm lateral to Bregma and for PFC recordings a craniotomy was performed over left frontal cortex at +1.78 mm anterior and +0.4 mm lateral to Bregma. The exposed skull was covered with a silicon elastomer (Body Double Fast, Smooth-on) until the first recording. All mice were given at least 2 days to recover from the surgery.

## Head-fixed virtual reality setup

The head-fixed virtual reality setup consisted of two rods that were screwed onto either side of the implanted head plate and fixated the mice on top of an air-supported spherical treadmill. The motion of the treadmill was recorded using an optical mouse and transformed into movement along a virtual linear track designed with the Blender rendering software. The virtual track was then projected through a mirror into a spherical screen surrounding the head-fixed animal on the treadmill (*Schmidt-Hieber and Häusser, 2013*; *Schmidt-Hieber, 2020*, https://github.com/neurodroid/gnoom). While in head fixation the animals received soy milk as reward which was delivered through a plastic spout that was positioned 0.5 cm anterior to the lower lip. Licks were detected with an infrared beam-break sensor that was positioned right in front of the spout.

All animals were slowly habituated to head fixation by placing them in the setup for at least 2 days of 3 × 10 min sessions during which they received about 50 rewards, totaling to about 0.2 ml of soy milk. During the habituation, we started to food restrict the animals to around 90 % of initial body weight to motivate better task performance.

In all cases, the food restricted animals started to lick off the soy milk reward reliably within the first six habituation sessions.

## Behavioral training

The AATC task required the animals to associate a 2-s-long CS+ sound with a droplet of soy milk reward (~5 μl), delivered after 1 s of silence, the so-called 'trace period' while ignoring a CS− control sound. We interleaved the CS presentations randomly every 30–45 s. For the two sounds, we choose a 3000 kHz continues pure tone and a 7000 kHz tone pulsating at 10 Hz. We counter balanced the CS+ and CS− sounds evenly between animals throughout the experiment.

As the main behavioral outcome measure, we detected the licks of the animals with an infrared beam-break sensor that was mounted in front of the reward spout. Each training session ended as soon as the animals received 50 rewards. We repeated the experiment for at least 10 days.

During these behavioral training sessions, the head-fixed animals could freely run on the linear track in the virtual reality which was otherwise not correlated with the AATC task.

We performed acute silicon probe recordings from the dorsal CA1 area and/or PFC of head-fixed mice during the first 2 days of the AATC task as well as from day 6 onwards for as long as we were able to achieve stable recordings. We then classified recording sessions from the first 2 days of training as prelearning (Pre) and recording sessions from day 6 onwards and with a significant increase in anticipatory licks as postlearning (Post).

## Acute electrophysiological recordings during AATC tasks

At the start of each recording session, we placed the mice in head fixation and removed the silicon elastomer cover to expose the skull. We then used a micromanipulator (Thorlabs) to acutely insert a 128-channel silicon probe into the middle of the previously prepared craniotomy above PFC and/or CA1. For PFC recordings, we then slowly lowered the recording electrode to –2.0 mm ventral to Bregma. For CA1 recordings, we continuously monitored the local field potential (1–600 Hz), ripple frequency signal (150–300 Hz), and spiking activity (600–2000 Hz) during the insertion process and tried to positioned our electrode in a way that the strongest ripple amplitude and spiking activity was 200–250 μm from the base of the probe and 470–520 μm from the tip. In this way, we were able to cover most of dorsoventral extend of CA1 with our recording electrode.

Electrophysiological signals were filtered between 1 and 6000 Hz, digitized at 30 kHz using 2 64 channel digitizing heads stages (RHD2164 Amplifier Board, Intan Technologies) and acquired with an open-ephys recording system. After each recording session, we retracted the silicon probe and placed a new silicon cover on the skull before releasing the animals back to their respective home cages.

## Behavioral data analysis

For every training session and each animal, we compared the change in lick rate between CS+ vs CS− trials. In short, for each trial we took the sum of all licks during the trace period and subtracted the sum of all licks during the baseline period (−1 s to onset of CS). We then computed a $t$-test between the change in lick rate for all CS+ vs all CS− trials and defined the animal to have learned (postlearning sessions) if this comparison showed a significant difference in lick rate between the two conditions.

## Neural data analysis

To identify single-unit activity, the raw voltage signal was automatically spike sorted with Kilosort (*Pachitariu et al., 2016*; https://github.com/cortex-lab/Kilosort) and then manually inspected and curated with the 'phy' gui (https://github.com/kwikteam/phy). All following analysis was performed using custom written MATLAB scripts (https://github.com/chanlukas/AATCstudy; *Klee, 2021*; copy archived at swh:1:rev:77c44a5dbb49f403f77333f29a524c88528a38e5).

## Single-cell responses

For each unit, we binned the single-cell spiking data (25 ms), smoothed the data with Gaussian-weighted moving average filter (25 bins), and computed the Peri-Stimulus-Time-Histograms (PSTHs) for CS+ and CS− trails.

To assess evoked responses to CS+ and CS− we calculated the $z$-scored firing rates in the first 350 ms (postlearning CS) poststimulus interval for each cell and compared population responses in this time window.

To assess lick-related activity, we first defined lick onset as the first lick after CS+ sound onset but before reward delivery that was followed by at least three licks within the next second. We then defined single cells to be lick responsive if the average firing rate during the time window around the first lick (−250 to +250 ms) was increased by at least one standard deviation or decreased by at least one standard deviation from pretrial baseline (−1000 – CS+ onset).

To assess trace period activity, we defined single cells to be trace responsive if the average $z$-scored firing rate during the 1- s trace period increased by at least one standard deviation or decreased by at least one standard deviation from baseline during either CS+ or CS− trials.

To quantify group differences, we performed a Kruskal–Wallis (pre- vs postlearning) using average $S$-scored responses during the trace period.

For analysis of learning-dependent changes in reward evoked activity, we calculated the $z$-scored firing rates during the reward response window (reward delivery − reward delivery +500 ms).

## Population rate vector analysis

To analyze the CA1 and PFC population response during the trace period, we first computed the average CS+ and CS− PSTHs for all simultaneously recorded nonlick cells in 25- ms bins for every session and smoothed the data with Gaussian-weighted moving average filter (25 bins). For visualization purposes, we then computed the first three principal components of the resulting matrices CS+ and CS− PSTHs and plotted the resulting three-dimensional vectors. To quantify the difference in population activity between CS+ and CS− trials, we calculated the Euclidean distance between CS+ and CS− in $n$-dimensional space ($n$ = number of simultaneously recorded nonlick cells) for each bin and averaged across sessions.

In order to predict trial identity by trace period population firing rates using a support vector machine classifier, we first calculated the trial-by-trial firing rates for all simultaneously recorded cells during the trace period. We then split all trials of a single recording session into 20 equal partitions and used 19 of these partitions to train the support vector machine classifier (fitcsvm function, MATLAB). We then tested the classifier performance on the 20th partition and repeated this process for all other partitions (20-fold crossvalidation). Classifier performance above chance was determined by comparing the average prediction accuracy across all sessions against chance level using a Wilcoxon sign rank test.

## RRR of CA1–PFC single-cell interactions

To investigate CA1–PFC interaction on the single-cell population level, we used RRR to assess how well the activity of a sampled population in one of the areas (source area) can be explain another (disjoint) sampled population in the same or in a separate target area, through a simplified, low-dimensional linear model (*Semedo et al., 2019*). Briefly, for a given session, we first subsample (without replacement) the population of each region in two equally sized sets of source and target neurons, so that all four sets had the same number of neurons. We then used a 10-fold crossvalidation scheme to compute the RRR (i.e., fitting the target population activity using the source population) using multiple rank values. The performance of each model was computed using the relative amount of variance explained by the model ($R^2$). We then selected the first model which had mean

performance within 1 standard error of the mean (SEM) of the best model, using its rank as the number of predictive dimensions (*Figure 4C*). This procedure was repeated for 10 different subsamples and the performance and number of predictive dimensions of each session was computed via averaging across subsamples. We also compared the performance of the full regression model (in which all the ranks were used) to control for different dimensionality of the RRRs in the two areas. In this particular case, we added L1 regularization, and chose the best model (highest average performance over crossvalidation) among different ridge parameter values and measured the MSE between estimated and real activity (*Figure 4—figure supplement 1*).

## Coherence analysis

To assess coherence between mPFC and CA1 during the AATC task, we first selected the CA1 recording channel with the strongest aSWR amplitude (see below) as well at the central channel of the mPFC recording electrodes and downsampled the raw voltage singles to 2000 Hz. Coherence was then analyzed with multitaper Fourier analysis (*Mitra and Pesaran, 1999*), using the Chronux MATLAB toolbox (http://www.chronux.org).

## Reactivation during aSWRs

In order to detected aSWRs during the intertrial periods of the AATC task, we first filtered the local field potential at the top half of our recording electrode (16 channels in total, 1 channel at each depth, 320 μm spread around CA1 cell layer) between 150 and 300 Hz, and used common average reference to exclude artifacts affecting all channels. We then identified the recording depth with the strongest average ripple power and used the ripple band signal on this channel for further analysis. For each recording session, we visually inspected the ripple band signal and manually set a low-cut threshold for ripple detection (100 μV in most cases) and a high-cut threshold for artifact rejection (500 μV in most cases). We furthermore, excluded threshold crossings within 200 ms of each other as well as within 200 ms of any licking activity.

To compute task-related cell assemblies in CA1 and PFC, we first binned the spikes of each trial (−1 to 4 s from stimulus onset) into 20- ms bins. Then, we used independent component analysis to find the coactivation patterns as described previously (*Lopes-dos-Santos et al., 2013*, *Figure 5—figure supplement 3*). The number of assemblies was defined by the eigenvalues of the crosscorrelation matrix that were above the analytical Marcenko–Pastur distribution (*Lopes-dos-Santos et al., 2013*). We then projected the neural activity onto each of the assembly patterns (using the same 20- ms bins, with overlap) and computed the mean assembly activity triggered by the stimulus, animal licks, and hippocampal SWR, normalizing it with the *z*-score transformation. Normalization was done using the *z*-score transformation, which in the case of stimulus-triggered assembly activity only used prestimulus period (of both CS+ and CS−) as baseline (−1000 to 0 s) and in case of ripples-triggered activity only used the period outside the ripple center (50- ms window) as baseline. At last, we defined the assembly reactivation window on SWR as the average *z*-scored assembly activity in the 50- ms window centered on the aSWR. We then divided each session in 10 equally long-trial blocks, and investigated the reactivation strength of positively (reactivation+) and negatively (reactivation−) modulated assemblies over the course of the session. Statistical comparisons between pre- and post (normalized) mean assembly activity were done using a Wilcoxon rank sum test, while comparisons of reactivation within a session was done comparing the reactivation in the first three and last blocks using a Wilcoxon signed-rank test. In *Figure 5D–E*, traces were smoothed using a 100 ms Gaussian window. Comparisons between CS+ and CS− stimuli were done using a Wilcoxon signed-rank test, while comparisons with baseline (prestimulus period) were done using a *t*-test.

## Trial-type encoding assembly reactivation during aSWR

For each assembly a trial-type modulation score (*Figure 6A*) was computed defined as the average assembly activation (in a given period) during CS+ trials minus the average activation during CS− trials. Similarly, a trial-period modulation score was defined as the average assembly activation during stimulus minus the average assembly activation during trace (*Figure 6—figure supplement 1A*). For trial-period modulation scores, only CS− trials were used. In *Figure 6B* and *Figure 6—figure supplement 1B*, only the assemblies with the lowest (CS+ suppressed), highest (CS− suppressed), and least (nonmodulated) score values of each session were chosen.

## CA1–PFC assembly coactivation during aSWR

First, to ensure that both areas had significant aSWRs reactivations, we computed the percentage of significant ripples. This was defined as the percentage of reactivations above two standard deviations (computed through the median absolute deviation) from the median. Then, to assess the coordination of CA1 and PFC assembly reactivation (during aSWRs), we counted how often each simultaneously recorded pair (one CA1 and one PFC assembly) reactivated together above the median in both areas and then compared the percentage of coincident high reactivations (quadrant 1; Q1) with the percentage of high and low reactivations in CA1 and PFC, respectively (quadrant 2; Q2; see *Figure 6*). A more conservative version of this analysis using the 5 % highest and lowest aSWRs reactivation to define the quadrant thresholds yield similar results (data not shown).

## Histology

At the end of each experiment, mice were perfused with 4% PFA and brain sections (100 µm) were examined with light microscopy to confirm electrode placement in CA1 and mPFC.

## Acknowledgements

Funding was provided by a German Studiensstiftung fellowship (to JK), by the Dutch NWA 'Bio-Art' project (to FPB), and by the NWO Top-grant no. 612.001.853 (to FPB).

## Additional information

### Funding

| Funder | Grant reference number | Author |
|---|---|---|
| Studienstiftung des Deutschen Volkes | | Jan L Klee |
| Nederlandse Organisatie voor Wetenschappelijk Onderzoek | 612.001.853 | Bryan C Souza |
| NWA 'Bio-Art' project | | Francesco P Battaglia |

The funders had no role in study design, data collection and interpretation, or the decision to submit the work for publication.

### Author contributions

Jan L Klee, Conceptualization, Data curation, Formal analysis, Funding acquisition, Investigation, Methodology, Project administration, Software, Validation, Visualization, Writing – original draft, Writing – review and editing; Bryan C Souza, Conceptualization, Data curation, Formal analysis, Methodology, Software, Validation, Visualization, Writing – original draft, Writing – review and editing; Francesco P Battaglia, Conceptualization, Formal analysis, Funding acquisition, Methodology, Project administration, Software, Supervision, Writing – original draft, Writing – review and editing

### Author ORCIDs

Jan L Klee http://orcid.org/0000-0003-4988-5682
Bryan C Souza http://orcid.org/0000-0002-1041-4624
Francesco P Battaglia http://orcid.org/0000-0003-3715-8875

### Ethics

This study was approved by the Central Commissie Dierproeven (CCD) in the Netherlands and conducted in accordance with the Experiments on Animals Act and the European Directive 2010/63/EU on animal research.

### Decision letter and Author response

Decision letter https://doi.org/10.7554/eLife.65456.sa1
Author response https://doi.org/10.7554/eLife.65456.sa2

## Additional files

### Supplementary files
• Transparent reporting form

### Data availability
All data are publicly available on the Donders Repository (https://doi.org/10.34973/hp7x-4241). Analysis scripts can be downloaded via GitHub (https://github.com/chanlukas/AATCstudy; copy archived at https://archive.softwareheritage.org/swh:1:rev:77c44a5dbb49f403f77333f29a524c88528a38e5).

The following dataset was generated:

| Author(s) | Year | Dataset title | Dataset URL | Database and Identifier |
|-----------|------|---------------|-------------|-------------------------|
| Klee JL, Souza BC, Battaglia FP | 2020 | Learning differentially shapes prefrontal and hippocampal activity | https://doi.org/10.34973/hp7x-4241 | Donders Repository, 10.34973/hp7x-4241 |

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
