## [Decision Letter]

**Acceptance summary:**

This paper is of interest to scientists using classical conditioning and others interested in prefrontal-hippocampal interactions in general. In particular, the reactivation of neuronal ensemble activity during non-spatial tasks is novel and fills a critical gap in knowledge.

**Decision letter after peer review:**

Thank you for submitting your article "Learning differentially shapes prefrontal and hippocampal activity patterns during classical conditioning" for consideration by *eLife*. Your article has been reviewed by 3 peer reviewers, including Kaori Takehara-Nishiuch as the Reviewing Editor and Reviewer #1, and the evaluation has been overseen by Laura Colgin as the Senior Editor.

Essential revisions:

1. The finding of reactivation during aSWRs in inter-trial intervals (ITI) provides the major novel advance provided by the manuscript. The authors conclude that this reactivation plays a role in learning the task. This claim requires further supporting evidence, including the demonstration of the relationship of reactivated cell assemblies with activity patterns described for CS+ and CS- stimuli as well as details about the rate and distribution of aSWRs over learning. For example, can the authors classify the detected cell assemblies into groups, each of which represents a specific task event (CS+, CS-, trace period, reward)? By running all subsequent analyses separately in each of these groups, it is possible to demonstrate how the timing and incidence of reactivation differ among these groups. Alternatively, similar to CA1 neurons, PFC neurons also change firing patterns depending on contextual information (see Hyman et al., 2012). Can the author identify "control" cell assemblies that co-fired irrespectively to task events by running the same assembly analysis with firings during the ITI period (excluding SWR events)? If the task-related information is preferentially reactivated, these control assemblies should be reactivated less frequently than the assemblies representing task events.

2. A few previous studies have reported prefrontal reactivation during behavior (Maggi et al., 2018; Shin et al., 2019; Kaefer et al. 2020) – only one is cited (Kaefer et al.), and not in the context of reactivation. Their analyses focused on the coordination of PFC and CA1 reactivation during aSWRs, which contrasts with the results in Figure 5 that demonstrate the assembly reactivation separately in each region. The authors need to address whether the reactivation is related or coordinated between the two regions and how this may change during learning. For example, do PFC and CA1 assemblies reactivate during the same aSWRs, which can potentially aid task-related plasticity and learning in the CA1-PFC circuit, or are they reactivated separately?

3. It is expected that ripples also occurred during the reward consumption period. The ripples here were within an active task phase and should yield more information relevant to the task. Did the authors look at the ripples during the reward consumption period?

4. The study is the first to simultaneously record from CA1 and PFC in trace conditioning. Despite this strength, the current form applies the same set of analyses to the two regions in parallel, leaving inter-region relationships and coordinations mostly unexplored. For example, is there any relationship or coordination between Trace-up and Trace-down neurons in PFC and CA1 reported in Figure 3, or is there any evidence of oscillatory coordination in the CA1-PFC circuit? At the minimum, cross-correlation analyses can be added to examine temporal relationships between the firing of PFC and CA1 neurons with distinct response sub-types.

5. In the analyses for examining evoked and trace period responses (Figures 1,2), the authors primarily use a subtractive measure (change in firing rate from baseline) and average across all neurons to infer sustained trace-period suppression in CA1 and enhancement in PFC for CS+ stimuli (Figure 1F-I, K-N). This population presumably consists of neurons with variable baseline rates and levels of responsiveness, as seen in the Z-scored figures (Figures 1E, J). Therefore, the same analysis can be done using the Z-scored measure, which can control variable baseline rates and avoid outlier contributions from neurons with high firing rates. The authors also need to confirm that the differing responses of overall suppression in CA1 and enhancement in PFC persist when using alternative measures.

6. Firing suppression in CA1 is interesting but expected given "rate-decreasing cells" reported in previous works with trace eyeblink conditioning (McEchron et al., 1999; Hattori et al., 2015). Even in the prelimbic cortex, about half of stimulus-responding cells decrease their firing rates in response to the CS (Takehara-Nishiuchi and McNaughton, 2008; Hattori et al., 2014). The authors need to relate their results to these existing findings and discuss a model explaining how, despite both Trace-up and Trace-down neurons, the overall effect is suppression in CA1 and enhancement in PFC. This can presumably occur due to the different strength of contributions of these two kinds of neuronal responses in the two regional populations.

7. In Figures 3and4, the authors used population vector distance to track the separation between the representations of two different sounds versus time. They all showed a sustained increase from the baseline. First, the distance was expressed as a Z-score. What distribution was the Z-score calculated against? Second, did the distance between the two sounds ever come back to the baseline level, e.g., close to 0, meaning not separable anymore? This could be a way to validate the method or check the stability of the recording.

8. As the authors wrote, traditionally sustained firing patterns are considered a neural correlate for the association of temporally discontinuous stimuli. However, recent studies using trace eyeblink conditioning (Modi et al., 2014; Pilkiw and Takehara-Nishiuchi, 2018) show that CA1 and the prelimbic region also contain cells transiently increase firing rates during a specific time segment and form a sequential firing pattern that bridges the trace period. These cells will not pass the authors' criterion for "trace-responsive" cells, which is based on the averaged firing rate in the entire 1-second delay period. Given the scarcity of the large-scale neural activity recording during explicitly non-spatial tasks, it is worth investigating whether or not the authors observe similar sequential patterns in either region.

9. The medial prefrontal cortex consists of several sub-regions, such as rostral anterior cingulate, prelimbic, and infralimbic cortex. In particular, Hattori et al. (2014) reported substantial differences in firing selectivity between the rostral cingulate and prelimbic area in trace eyeblink conditioning. Therefore, the authors need to show histology and report which sub-region was the primary source of recordings.

10. Another concern is how the authors ensure the stability of their recordings. Acute recording is less table than chronic recording. Is there any analysis or method for the authors to only include the stably recorded cells in the analyses?

11. In several sections (e.g., Line 121), the authors first used ANOVA and then follow-up with rank-sum tests. Please check the normality of the data and use either parametric or non-parametric tests.

[Editors' note: further revisions were suggested prior to acceptance, as described below.]

Thank you for submitting your article "Learning differentially shapes prefrontal and hippocampal activity patterns during classical conditioning" for consideration by *eLife*. Your article has been reviewed by 3 peer reviewers, including Kaori Takehara-Nishiuch as the Reviewing Editor and Reviewer #1, and the evaluation has been overseen by Laura Colgin as the Senior Editor.

Essential revisions:

All reviewers appreciated the authors' effort in conducting additional analyses to strengthen their claims further. In particular, the new results successfully tighten the link between learning and detected neural activity patterns. Although Reviewers 1 and 2 thought their concerns were fully addressed, Reviewer 3 raised several concerns. Among them, we agree that the third comment is critical and invite the authors to include additional discussion on the following points:

1) The role of aSWR reactivation in non-spatial associative learning.

2) The uncorrelated reactivation between CA1 and mPFC in light of contradicting findings by other groups

In addition, please address editorial suggestions/concerns raised by all reviewers.

*Reviewer #3 (Recommendations for the authors):*

The authors have addressed several concerns in the revised manuscript. In particular, they have clarified key analyses, and added new analyses examining coordination of CA1-PFC ensembles during the trace conditioning task. The revised manuscript does a better job of addressing of how physiological phenomena contribute to learning of the task. There are still a few areas that remain unaddressed and additional clarification can be added.

1) The authors add LFP coherence and reduced rank regression (RRR) analyses to examine CA1-PFC interactions during trace conditioning (Figure 4). Here, it will be useful to show examples of raw LFP traces in CA1 and PFC from which the coherence is derived.

The RRR analyses is informative about directionality of interactions, and similar CA1-PFC interactions are observed across task periods, suggesting generality of interactions unrelated to the task. The question of whether the different cell type responses, Trace-up and Trace-down neurons, across CA1 and PFC regions are correlated or not is not addressed. Since these response types are shown to be task-specific in Figure 2, examining correlations for specific neuronal response types across regions has the potential to reveal an interaction, and is a missed opportunity.

2) The authors also choose not to examine possibility of transient increases in neuronal activity and sequential patterns, which can also drive the population vector trajectories shown in Figure 3.

3) The authors have added to the reactivation analyses to examine what kinds of information is reactivated in CA1 and PFC during aSWRs in the inter-trial periods. They report reactivation of CS+ specific information in CA1, and non-specific reactivation of all assemblies in PFC that increases over learning. The authors also report that reactivation across regions remains uncorrelated.

In the light of these new findings, the authors can provide additional interpretation regarding the possible role of aSWR reactivation in the learning of this task. It seems that aSWR reactivation is only beneficial for within-region changes during learning? Although role of intermediate regions such as entorhinal cortex is speculated, reactivation in these regions can also be independent.

The finding of uncorrelated reactivation and associated analyses can also use additional explanation. As reported in Figure 6D, the proportion of reactivation in each region was compared against the rate of joint reactivation across CA1-PFC assemblies, with the finding that joint reactivation was no more likely than independent reactivation. Here, it is important to report the probability of significant assembly reactivation (reactivation rate per aSWR) in CA1 and PFC. If these probabilities are low, then it is possible that data limitations constrain the statistical probability of observing joint reactivation across assemblies. Although Figure 5 shows the strength of assembly reactivation, the rate of this reactivation per aSWR is not clear.

Finally, given that PFC reactivation is detected during hippocampal aSWRs, which drive synchronous reactivation of prefrontal neurons (Nishimura et al., Cell Reports, 2021), it will be useful to hear the authors' interpretation of why prefrontal reactivation may be uncorrelated, the possible function of independent reactivation in task learning, and if perhaps post-task sleep SWRs can have a role in driving synchronous CA1-cortical reactivation for learning.

---

## [Author Response]

Essential revisions:1. The finding of reactivation during aSWRs in inter-trial intervals (ITI) provides the major novel advance provided by the manuscript. The authors conclude that this reactivation plays a role in learning the task. This claim requires further supporting evidence, including the demonstration of the relationship of reactivated cell assemblies with activity patterns described for CS+ and CS- stimuli as well as details about the rate and distribution of aSWRs over learning. For example, can the authors classify the detected cell assemblies into groups, each of which represents a specific task event (CS+, CS-, trace period, reward)? By running all subsequent analyses separately in each of these groups, it is possible to demonstrate how the timing and incidence of reactivation differ among these groups.

We thank the reviewers for this insightful comment and we agree that reactivation during aSWR is an important, novel result that invites further investigations.

To find evidence for reactivation of task event specific information during aSWR (see new Figure 6), we first computed the average activity of each assembly for CS+ or CS- stimuli and then computed a trial-type modulation score, defined as the average assembly activation during CS+ trials subtracted by the average activation during CS- trials (stimulus and trace periods; Figure 6A). Then, for each session, we selected the most positively modulated (i.e., CS+ activity higher than CS-), the most negatively modulated (i.e., CS- activity higher than CS+), and the least modulated “control” assembly. Because we found that most assemblies in CA1 are suppressed during the stimulus (as are the firing rates, Figure 1 F and H), we term assemblies by whether they were more suppressed during CS+ or CS- stimuli.

We found that, for CA1, CS+ suppressed assemblies in stimulus and trace periods (i.e., assemblies that are more suppressed during CS+ trials than CS- trials), were more reactivated during aSWRs (Figure 6B) compared to CS- suppressed assemblies and control, non-modulated assemblies. This effect was confirmed by a negative correlation between the trial-type modulation score and the average aSWR reactivation of each assembly (Figure 6C). CA1 therefore preferentially reactivates assemblies that encode task specific information. In PFC we did not observe preferential reactivation of CS+ or CS- specific assemblies (Supp. Figure 6).

Alternatively, similar to CA1 neurons, PFC neurons also change firing patterns depending on contextual information (see Hyman et al., 2012). Can the author identify "control" cell assemblies that co-fired irrespectively to task events by running the same assembly analysis with firings during the ITI period (excluding SWR events)? If the task-related information is preferentially reactivated, these control assemblies should be reactivated less frequently than the assemblies representing task events.

In the revised version of the manuscript, we included a new control analysis in which we do as suggested by the reviewer. We detected the assemblies during the inter-trial intervals (excluding aSWRs events) and investigated those assembly reactivations during aSWR (Supp. Figure 5.1).

We find that reactivation of assemblies defined during the intertrial interval is similar to reactivation of assemblies defined during the trial. However, groups of cells that are coactive during the trial will often also be coactive during the intertrial interval and vice versa which makes a comparison between “trial” and “ITI” assembly reactivation hard to interpret.

In contrast, our findings in new Figure 6 clearly highlight that assemblies that specifically respond to CS+ stimuli with suppression, are more re-activated than assemblies that are not modulated by CS+ stimuli and, in our view, provide solid evidence that in CA1 CS+ specific assemblies are preferentially reactivated.

On the other hand, neither analysis produced evidence of CS specific reactivation in PFC. We therefore conclude that increased reactivation from pre to post learning session is not CS specific and instead, general task context dependent and reward related (See Figure 5 E) assemblies are reactivated during aSWR in PFC.

2. A few previous studies have reported prefrontal reactivation during behavior (Maggi et al., 2018; Shin et al., 2019; Kaefer et al. 2020) – only one is cited (Kaefer et al.), and not in the context of reactivation. Their analyses focused on the coordination of PFC and CA1 reactivation during aSWRs, which contrasts with the results in Figure 5 that demonstrate the assembly reactivation separately in each region. The authors need to address whether the reactivation is related or coordinated between the two regions and how this may change during learning. For example, do PFC and CA1 assemblies reactivate during the same aSWRs, which can potentially aid task-related plasticity and learning in the CA1-PFC circuit, or are they reactivated separately?

Indeed, reactivation in PFC during aSWR is likely an important phenomenon that underlies learning and memory. The literature cited by the reviewer shows that reactivation in PFC has previously been observed in spatial memory tasks and we now emphasize this in the manuscript. However, whether reactivation is coordinated between CA1 and PFC is still a matter of debate (Shin et al. 2019 find coordination, Kaefer et al. 2020 do not). To check whether CA1 and PFC assembly reactivation is coordinated during aSWR during trace conditioning, we counted how often each simultaneously recorded assembly pair (1 CA1 and 1 PFC assembly) reactivated together above the median in both areas and then compared the percentage of coincident high reactivations (Q1) with the percentage of high and low reactivations in CA1 and PFC respectively (Q2). (Figure 6D). In line with previous results (Kaefer et al. 2020), we did not find evidence for increased co-reactivation of CA1 and PFC assemblies during aSWRs. In combination, with analysis of single cell coordination between both areas during the trials (see below), this suggests that CA1 and PFC process information during trace-conditioning in parallel rather than in series or by means of an intermediate structure (e.g., entorhinal cortex). We now highlight this result and interpretation in a revised version of our discussion.

3. It is expected that ripples also occurred during the reward consumption period. The ripples here were within an active task phase and should yield more information relevant to the task. Did the authors look at the ripples during the reward consumption period?

We thank the reviewers for raising an interesting point. Indeed, aSWR are elevated at reward locations in spatial navigation studies (Singer and Frank, 2009). However, differences in aSWR rate at the exact moments of chewing/licking behavior compared to non-chewing/licking at the same reward location are difficult to detect in freely moving animals and, to the best of our knowledge, have not been previously reported. Furthermore, aSWR in response to behaviorally relevant sounds have not been extensively described before.

In our head-fixed trace-conditioning experiments, we were able to precisely detect instances of licking for reward as well as CS+ and CS- sound presentations and align aSWR rate to these events.

To our surprise, we observed a marked decrease in the rate of aSWR during conditioning trials and almost a complete absence of aSWR while mice were licking for the reward. While conceptually interesting, we were therefore not able to perform the analysis suggested by the reviewer above because of a lack of aSWR during the precise reward consumption period.

However, the baseline rate of aSWR occurrence during the ITIs in our task was similar to what has previously been reported at familiar reward locations (Singer Frank, Neuron 2009).

To emphasize these points, we included a Supplementary Figure 5.2 that shows the distribution of aSWR during trials, sessions and learning stages.

4. The study is the first to simultaneously record from CA1 and PFC in trace conditioning. Despite this strength, the current form applies the same set of analyses to the two regions in parallel, leaving inter-region relationships and coordinations mostly unexplored. For example, is there any relationship or coordination between Trace-up and Trace-down neurons in PFC and CA1 reported in Figure 3, or is there any evidence of oscillatory coordination in the CA1-PFC circuit? At the minimum, cross-correlation analyses can be added to examine temporal relationships between the firing of PFC and CA1 neurons with distinct response sub-types.

We thank the reviewers for this insightful comment. In fact, we originally set up parallel recordings from CA1 and PFC because we hypothesized that these two areas which have been shown to interact on the level of single cells and field potentials in various other memory tasks would also show coordinated activity during trace conditioning. We are excited that the reviewers share our interest in this topic and decided to include an additional main Figure 4 in the manuscript.

Heightened oscillatory coherence between CA1 and PFC has previously been described during spatial working memory tasks. We therefore decided to check for oscillatory coherence between CA1 and PFC. We found a significant increase in CA1-PFC coherence during the trace-period after CS+ compared to CS- trials (12-140 Hz) (Figure 4 A).

CA1 – PFC coherence during spatial working memory tasks was reported predominantly in the Theta range (4-12Hz). In our experiments mice were mostly stationary (see Supp. Figure 3.2). Expectedly, we therefore did not detect strong theta nor a specific peak in theta coherence.

We next checked whether we could find evidence for single cell coordination during conditioning trials (Figure 4B). To this end we computed a reduced-rank regression (RRR) to assess how well the activity of a sampled population in one of the areas (source area) can be used to explain the activity of another (disjoint) sampled population in the same area or in a separate target area, through a simplified, low-dimensional linear model. We then used cross-validation to estimate the optimal dimensionality (rank) of each RRR and its performance (R^2^) (Semedo et al., 2019). We observed that CA1 ensemble activity at baseline could be used to predict other individual neurons firing rates in CA1 just as well as the firing rates of neurons in PFC. The PFC ensemble on the other hand was much better at predicting the firing rates of other PFC neurons over neurons in CA1 which indicates a directionality of information flow between both areas at baseline.

However, cross-area predictability of firing rates did not change significantly when we compared baseline levels to any of the different trial stages (Stim, Trace or Reward), indicating that communication between the two areas might depend on more general, context dependent variables instead of specific conditioning trial specific information and that both areas process trial type specific information in parallel rather than in series. We had similar results when comparing the performance of the full-rank model across different periods (Ridge-regression with L1 regularization; Supp. Figure 6) We updated our result section, figures and discussion to present these new findings accordingly.

5. In the analyses for examining evoked and trace period responses (Figures 1,2), the authors primarily use a subtractive measure (change in firing rate from baseline) and average across all neurons to infer sustained trace-period suppression in CA1 and enhancement in PFC for CS+ stimuli (Figure 1F-I, K-N). This population presumably consists of neurons with variable baseline rates and levels of responsiveness, as seen in the Z-scored figures (Figures 1E, J). Therefore, the same analysis can be done using the Z-scored measure, which can control variable baseline rates and avoid outlier contributions from neurons with high firing rates. The authors also need to confirm that the differing responses of overall suppression in CA1 and enhancement in PFC persist when using alternative measures.

We thank the reviewers for this insightful comment and we agree that representing changes in firing rates as Z-scores consistently throughout the text improves the interpretability of our data. We therefore updated Figures 1 and 2 as well as the corresponding paragraphs in the text accordingly. Overall, we had very similar results, however, sound evoked firing rates in CA1 post-learning were found to be not significantly different from each other when comparing Z-scored firing rates (Figure 1 G).

6. Firing suppression in CA1 is interesting but expected given "rate-decreasing cells" reported in previous works with trace eyeblink conditioning (McEchron et al., 1999; Hattori et al., 2015). Even in the prelimbic cortex, about half of stimulus-responding cells decrease their firing rates in response to the CS (Takehara-Nishiuchi and McNaughton, 2008; Hattori et al., 2014). The authors need to relate their results to these existing findings and discuss a model explaining how, despite both Trace-up and Trace-down neurons, the overall effect is suppression in CA1 and enhancement in PFC. This can presumably occur due to the different strength of contributions of these two kinds of neuronal responses in the two regional populations.

We thank the reviewer for bringing up this important point and we have consequently elaborated on the distribution of rate increasing and decreasing cells across both areas as well as on how the numbers that we find in our study relate to previous work by adding an additional supplementary figure (Sup. Figure 2) and by expanding on this point in our discussion.

We also substantially revised the first half of the Discussion section to properly contrast our results to previous findings in the trace-conditioning field:

“This study characterizes changes in neural activity in the CA1-PFC network while mice learn to use predictive sounds to anticipate future rewards. We show that activity in both areas is strongly shaped by learning and that task specific information is reactivated in a complex pattern across CA1 and PFC during aSWR.

While CA1 and PFC are highly active during aversive eyeblink trace conditioning, evidence for a similar involvement during appetitive trace-conditioning had been missing. In fact, several previous studies have pointed out differences in the mechanisms underlying both types of learning (McEchron et al., 1999; Pezze et al., 2017; Thibaudeau et al., 2007).

Despite these differences, we found that single cells in CA1 and mPFC during appetitive trace-conditioning behave similarly to what had previously been reported during aversive trace-conditioning. Both areas display long lasting sustained activity that bridges the temporal gap between CS+ offset and reward delivery. In both areas, these sustained responses are composed of a mix of Trace-Up and Trace-down cells, i.e. cells that display sustained excitation and inhibition respectively. In CA1, higher numbers and stronger inhibition of Trace-Down cells result in overall suppression of the entire area during the trace period, while in PFC higher numbers and stronger activation of Trace-Up cells resulted in overall excitation.

Similar to our study, abundant Trace-Down like responses and spares Trace-Up like responses have been described during aversive trace-conditioning in CA1(Hattori et al., 2015; McEchron and Disterhoft, 1997). This distinct pattern of mostly inhibition mixed with sparse excitation has been hypothesized to increase the signal to noise ratio to more efficiently propagate the signal of Trace-Up cells to downstream areas (Hattori et al., 2015) Yet, it is also conceivable that Trace-down cells participate in an independent form of coding. Inhibition in CA1 might for example play an active role in suppressing well expected incoming stimuli, i.e. reward delivery (Bastos et al., 2012; Rummell et al., 2016; Stachenfeld et al., 2017).

In PFC, responses during appetitive trace-conditioning are also similar to what has previously been found during aversive trace-conditioning. Specifically, higher numbers and stronger excitation of Trace-Up cells have also been found in rat PFC and parts of rabbit PFC during aversive trace-conditioning (Hattori et al., 2014; Takehara-Nishiuchi and McNaughton, 2008). A learning dependent reduction in responses to CS- like pseudo conditioning stimuli have also previously been described in PFC (Hattori et al., 2014; Takehara-Nishiuchi and McNaughton, 2008; Weiss and Disterhoft, 2011).

In combination, this suggests that sparse excitation with strong surrounding inhibition in CA1 and mostly excitation in PFC are two general coding principles employed to bridge the temporal gap between a salient cue and a behaviorally relevant event, independently of the appetitive or aversive nature of the event and the specific anticipatory action that it requires.“

7. In Figures 3and4, the authors used population vector distance to track the separation between the representations of two different sounds versus time. They all showed a sustained increase from the baseline. First, the distance was expressed as a Z-score. What distribution was the Z-score calculated against? Second, did the distance between the two sounds ever come back to the baseline level, e.g., close to 0, meaning not separable anymore? This could be a way to validate the method or check the stability of the recording.

We thank the reviewers for this thoughtful comment which motivated us to include an additional supplementary Figure 3.1 that highlights the changes in firing rate and population vector distance over a longer time period (25s) after the conditioning trials.

In our main data set, we found that firing rates and population rate vector distance 20 s after reward delivery are back at baseline levels. This closely matches the time course within which average single cell activity settles back to baseline levels and is well within the inter trial interval of 30-45 s.

The Z-score of the population rate vector distance was calculated using the mean and standard deviation of the averaged population rate vector distances at baseline (-5s – 0) before sound onset.

We furthermore noticed that the comparison between CS+ (a) and CS+(b) did not reach significance anymore when using raw (not Z-scored) single cell PSTHs to calculate the population rate vector distances. We therefore excluded previous Figure 4 from the paper.

8. As the authors wrote, traditionally sustained firing patterns are considered a neural correlate for the association of temporally discontinuous stimuli. However, recent studies using trace eyeblink conditioning (Modi et al., 2014; Pilkiw and Takehara-Nishiuchi, 2018) show that CA1 and the prelimbic region also contain cells transiently increase firing rates during a specific time segment and form a sequential firing pattern that bridges the trace period. These cells will not pass the authors' criterion for "trace-responsive" cells, which is based on the averaged firing rate in the entire 1-second delay period. Given the scarcity of the large-scale neural activity recording during explicitly non-spatial tasks, it is worth investigating whether or not the authors observe similar sequential patterns in either region.

Indeed, neural ensemble patterns in this task are extremely complex, as highlighted by the literature cited by the reviewer. We are now analyzing our data looking for sequential activations and other patterns with advanced statistics developed with our collaborators. We believe that this work falls outside the scope of the present paper, which concentrates more on firing rate modulations, and will be the subject of a dedicated additional manuscript that is currently in preparation.

9. The medial prefrontal cortex consists of several sub-regions, such as rostral anterior cingulate, prelimbic, and infralimbic cortex. In particular, Hattori et al. (2014) reported substantial differences in firing selectivity between the rostral cingulate and prelimbic area in trace eyeblink conditioning. Therefore, the authors need to show histology and report which sub-region was the primary source of recordings.

The specific recording position within the frontal cortex is indeed a crucial factor for the interpretation of our data and we thank the reviewer for pointing this out. We included a representative histological image and a schematic of the estimated electrode position in 6 animals in PFC and 4 animals in CA1 as a Supplementary Figure 1**.**

We set out to record from deep layers of prelimbic cortex at 1.8mm anterior and 2.4mm ventral from Bregma. Our recordings were performed with two prototype “Neuroseeker” 128 channel silicon probes that did not allow us to verify the exact recording position with electrolytic lesions. However, we managed to reconstruct the electrode position in 6 out of 17 animals in which the electrode track was visible under the light microscope. In all 6 animals, the reconstructed electrode position was found slightly more anterior than planned at 1.9mm – 2.1mm anterior to Bregma at the correct depth of 2.4mm at the intersection between Prelimbic, Infralimbic and Medial Orbitofrontal cortex. The electrophysiological signature of CA1 allowed us to guide our recording electrodes to the correct position in dorsal CA1.

10. Another concern is how the authors ensure the stability of their recordings. Acute recording is less table than chronic recording. Is there any analysis or method for the authors to only include the stably recorded cells in the analyses?

We thank the reviewers for bringing up this important point. We did take great care to only including stably recorded cells in the analysis and we will shortly describe our approach below. However, we do want to point out that most of the analysis in this manuscript does not describe changes in neural activity over the course of an individual session but rather relies on averaging across randomly interleaved CS+ or CS- trials. We assume that for this analysis, cells fading in or out, could only “water down” any effect because these cells would be unresponsive in some trials (however, this should affect randomly interleaved CS+ and CS- trials evenly).

In general, to identify single unit activity, the raw voltage signal was automatically spike sorted with Kilosort (Pachitariu et al., 2016) (https://github.com/cortex-lab/Kilosort) and then manually inspected and curated with the ‘phy’ gui (https://github.com/kwikteam/phy).

During this step we took great care to only include stable units for further analysis.

To further verify the stability of our recordings, we split each recording session into 3 blocks and compared the average firing rates as well as the change in firing rate of all cells from the first to the last third of each session (See Author response image 1) .

**Author response image 1. sa2fig1:** 

We found no significant difference in firing rates between early and late periods in each session (Wilcoxon Ranksum p=.958). In addition, we found that 97% of all 3514 cells changed their firing rate by less than 10% and less than 0.5% of cells changed their firing rate by more than 30%.Based on these results, we feel confident that our recordings were stable and did not significantly affect our results.

11. In several sections (e.g., Line 121), the authors first used ANOVA and then follow-up with rank-sum tests. Please check the normality of the data and use either parametric or non-parametric tests.

We thank the reviewers for pointing this inconsistency out to us. Peak sound evoked responses in our data set were not normally distributed in CA1 (Kolmogorov-Smirnov Test p<0.001) and PFC (Kolmogorov-Smirnov Test p<0.001). We therefore resorted to performing a Kruskal-Valis nonparametric ANOVA and found significant group differences for peak sound evoked responses in CA1 (F(3,3267)=23.64, p<0.001) and PFC (F(3,4413)=21.49, p<0.001). Post-hoc Wilcoxon Rank Sum Tests revealed that peak CS+ and CS- sound evoked responses increased from Pre to Post learning in CA1 (Wilcoxon Rank Sum Test for CS+, p<0.001 and CS-, p=0.008). In PFC peak evoked responses to CS+ sounds did not change significantly but peak evoked responses to CS- stimuli decreased significantly (Wilcoxon Rank Sum Test for CS+, p=0.2 and CS-, p<0.001).

We updated the respective paragraph in the manuscript.

[Editors' note: further revisions were suggested prior to acceptance, as described below.]

Essential revisions:All reviewers appreciated the authors' effort in conducting additional analyses to strengthen their claims further. In particular, the new results successfully tighten the link between learning and detected neural activity patterns. Although Reviewers 1 and 2 thought their concerns were fully addressed, Reviewer 3 raised several concerns. Among them, we agree that the third comment is critical and invite the authors to include additional discussion on the following points:1) The role of aSWR reactivation in non-spatial associative learning.2) The uncorrelated reactivation between CA1 and mPFC in light of contradicting findings by other groups.

We thank the reviewers for raising these two important points. We decided to address them together in a new section of the discussion:

“Surprisingly, we did not observe that assemblies, derived from activity during conditioning trials in CA1 and PFC, significantly co-reactivated during aSWR.

Several studies previously reported single cells in CA1 and PFC with similar spatial firing fields to also be strongly correlated during aSWRs and that synaptic inputs to individual mPFC cells increased if CA1 replay was more coordinated (Nishimura et al., 2021; Shin et al., 2019; Tang et al., 2017). However, on the population level, CA1 and mPFC reactivation of specific spatial trajectories has been found to occur independently (Kaefer et al., 2020). Moreover, reactivation of spatial sequences in CA1 and other cortical areas, specifically the entorhinal cortex has been shown to occur independently as well (O’Neill et al., 2017).

One way to reconcile these findings is that the coordination of aSWRs reactivation within the CA1-PFC circuit might depend on task structure. If animals have to follow rules that are based on specific sequences of behavioral events, e.g., in spatial alternation tasks (Shin et al., 2019), replay of sequences of events in CA1 might drive the activation of cells or cell assemblies in PFC that encode the appropriate behavioral response to those sequences (Buzsáki and Tingley, 2018). If the task structure instead “only” requires stimulus response mappings, as in our experiment and visually guided spatial navigation experiments (Kaefer et al., 2020), PFC might not rely on additional information from CA1 and reactivation remains independent. However, coordinated aSWRs reactivation in CA1 and PFC might happen robustly in non-spatial tasks if the tasks require the animals to learn sequences (Cabral et al., 2014; Rondi-Reig et al., 2001; Terada et al., 2017).

Lastly, it would be intriguing to know how SWRs reactivation of task relevant information during classical conditioning depends on the current state of the animal. A previous study reported significant differences in coordinated CA1-PFC reactivation between wakefulness and sleep (Tang et al., 2021). Yet evidence for sleep SWR reactivation of nonspatial information is still lacking. This further highlights the importance to study SWR reactivation with a battery of different behavioral task and across behavioral states that can help to disentangle the exact content and relevance of replay events for learning and behavior in the future. "

In addition, please address editorial suggestions/concerns raised by all reviewers.

We addressed the additional comments.

Reviewer #3 (Recommendations for the authors):The authors have addressed several concerns in the revised manuscript. In particular, they have clarified key analyses, and added new analyses examining coordination of CA1-PFC ensembles during the trace conditioning task. The revised manuscript does a better job of addressing of how physiological phenomena contribute to learning of the task. There are still a few areas that remain unaddressed and additional clarification can be added.

We thank reviewer 3 for this positive feedback and the helpful suggestions for further improvement.

1) The authors add LFP coherence and reduced rank regression (RRR) analyses to examine CA1-PFC interactions during trace conditioning (Figure 4). Here, it will be useful to show examples of raw LFP traces in CA1 and PFC from which the coherence is derived.

We fully agree that examples of raw data are useful for the interpretation of our data. We therefore revised Figure 4 and added a new panel A which shows simultaneously recorded LFP and single unit activity from CA1 and PFC during one CS+ conditioning trial.

Furthermore, we decided to add example traces of simultaneously recorded LFP and single unit activity during aSWR to Figure 5—figure supplement 2 A.

The RRR analyses is informative about directionality of interactions, and similar CA1-PFC interactions are observed across task periods, suggesting generality of interactions unrelated to the task. The question of whether the different cell type responses, Trace-up and Trace-down neurons, across CA1 and PFC regions are correlated or not is not addressed. Since these response types are shown to be task-specific in Figure 2, examining correlations for specific neuronal response types across regions has the potential to reveal an interaction, and is a missed opportunity.

We thank the reviewer for pointing out this interesting avenue for further analysis. We completely agree with this reasoning but decided that as a first approach to find coordinated activity between both areas, we wanted to use a measure that is unbiased and does not rely on a previous classification into trace-up and trace-down cells. However, as pointed out below, we are currently working on a second manuscript and are approaching these questions with new analysis methods which we feel like fall beyond the scope of this manuscript.

2) The authors also choose not to examine possibility of transient increases in neuronal activity and sequential patterns, which can also drive the population vector trajectories shown in Figure 3.

We thank the reviewer for this insight full comment and we absolutely share the interest in sequential firing patterns during conditioning in CA1 and PFC. In fact, we are now analyzing our data looking for sequential activations and other patterns with advanced statistics developed with our collaborators. We believe that this work falls outside the scope of the present paper, which concentrates more on firing rate modulations, and will be the subject of a dedicated additional manuscript that is currently in preparation.

3) The authors have added to the reactivation analyses to examine what kinds of information is reactivated in CA1 and PFC during aSWRs in the inter-trial periods. They report reactivation of CS+ specific information in CA1, and non-specific reactivation of all assemblies in PFC that increases over learning. The authors also report that reactivation across regions remains uncorrelated.In the light of these new findings, the authors can provide additional interpretation regarding the possible role of aSWR reactivation in the learning of this task. It seems that aSWR reactivation is only beneficial for within-region changes during learning? Although role of intermediate regions such as entorhinal cortex is speculated, reactivation in these regions can also be independent.

We thank the reviewer for raising these important points. We decided to address them together with the reviewers finally comment on our view on how to reconcile our findings with previous studies that reported coordinated activity in CA1 and PFC during aSWRs. We added this paragraph to our original discussion:

“Surprisingly, we did not observe that assemblies, derived from activity during conditioning trials in CA1 and PFC, significantly co-reactivated during aSWR.

Several studies previously reported single cells in CA1 and PFC with similar spatial firing fields to also be strongly correlated during aSWRs and that synaptic inputs to individual mPFC cells increased if CA1 replay was more coordinated (Nishimura et al., 2021; Shin et al., 2019; Tang et al., 2017). However, on the population level, CA1 and mPFC reactivation of specific spatial trajectories has been found to occur independently (Kaefer et al., 2020). Moreover, reactivation of spatial sequences in CA1 and other cortical areas, specifically the entorhinal cortex has been shown to occur independently as well (O’Neill et al., 2017).

One way to reconcile these findings is that the coordination of aSWRs reactivation within the CA1-PFC circuit might depend on task structure. If animals have to follow rules that are based on specific sequences of behavioral events, e.g., in spatial alternation tasks (Shin et al., 2019), replay of sequences of events in CA1 might drive the activation of cells or cell assemblies in PFC that encode the appropriate behavioral response to those sequences (Buzsáki and Tingley, 2018). If the task structure instead “only” requires stimulus response mappings, as in our experiment and visually guided spatial navigation experiments (Kaefer et al., 2020), PFC might not rely on additional information from CA1 and reactivation remains independent. However, coordinated aSWRs reactivation in CA1 and PFC might happen robustly in non-spatial tasks if the tasks require the animals to learn sequences (Cabral et al., 2014; Rondi-Reig et al., 2001; Terada et al., 2017).

Lastly, it would be intriguing to know how SWRs reactivation of task relevant information during classical conditioning depends on the current state of the animal. A previous study reported significant differences in coordinated CA1-PFC reactivation between wakefulness and sleep (Tang et al., 2021). Yet evidence for sleep SWR reactivation of nonspatial information is still lacking. This further highlights the importance to study SWR reactivation with a battery of different behavioral task and across behavioral states that can help to disentangle the exact content and relevance of replay events for learning and behavior in the future."

The finding of uncorrelated reactivation and associated analyses can also use additional explanation. As reported in Figure 6D, the proportion of reactivation in each region was compared against the rate of joint reactivation across CA1-PFC assemblies, with the finding that joint reactivation was no more likely than independent reactivation. Here, it is important to report the probability of significant assembly reactivation (reactivation rate per aSWR) in CA1 and PFC. If these probabilities are low, then it is possible that data limitations constrain the statistical probability of observing joint reactivation across assemblies. Although Figure 5 shows the strength of assembly reactivation, the rate of this reactivation per aSWR is not clear.

We thank the reviewer for pointing this issue and we now address the aSWRs joint reactivations in more detail. First. we computed the rate of significant aSWRs reactivation as suggested by the reviewer. To do so, we used a threshold of 2sd above the median (s.d. was estimated using the median absolute deviation to minimize the effects of outliers). We found that ~5% of the reactivations were above this threshold. This is now shown in Figure 6—figure supplement 1.

We then recomputed the proportion of joint reactivations (as in Figure 6D), but using the quadrants defined now by the 5% of highest and lowest reactivations. We found that the percentage of joint CA1/PFC reactivations falling in the 5% highest on both areas was not different than the other three quadrants (i.e., 5% lowest in CA1 and 5% lowest in PFC; 5% highest in CA1 and 5% lowest in PFC; and the opposite).

We show that in the figure above (right), but choose not to include it in the revised manuscript to avoid redundancy with Figure 6 D.

Finally, given that PFC reactivation is detected during hippocampal aSWRs, which drive synchronous reactivation of prefrontal neurons (Nishimura et al., Cell Reports, 2021), it will be useful to hear the authors' interpretation of why prefrontal reactivation may be uncorrelated, the possible function of independent reactivation in task learning, and if perhaps post-task sleep SWRs can have a role in driving synchronous CA1-cortical reactivation for learning.

We appreciate the reviewer’s comment and addressed this in our revised discussion posted above.